# Engineering multifunctional bactericidal nanofibers for abdominal hernia repair

Samson Afewerki [1,2✉], Nicole Bassous[3], Samarah Vargas Harb [3,4], Marcus Alexandre F. Corat [5], Sushila Maharjan[1,2], Guillermo U. Ruiz-Esparza[1,2], Mirian M. M. de Paula[5], Thomas J. Webster[3], Carla Roberta Tim[6], Bartolomeu Cruz Viana[7,8], Danquan Wang[3], Xichi Wang[1,2], Fernanda Roberta Marciano [3,8] & Anderson Oliveira Lobo [1,2,7,9✉]

The engineering of multifunctional surgical bactericidal nanofibers with inherent suitable mechanical and biological properties, through facile and cheap fabrication technology, is a great challenge. Moreover, hernia, which is when organ is pushed through an opening in the muscle or adjacent tissue due to damage of tissue structure or function, is a dire clinical challenge that currently needs surgery for recovery. Nevertheless, post-surgical hernia complications, like infection, fibrosis, tissue adhesions, scaffold rejection, inflammation, and recurrence still remain important clinical problems. Herein, through an integrated electro-spinning, plasma treatment and direct surface modification strategy, multifunctional bactericidal nanofibers were engineered showing optimal properties for hernia repair. The nanofibers displayed good bactericidal activity, low inflammatory response, good biodegradation, as well as optimal collagen-, stress fiber- and blood vessel formation and associated tissue ingrowth in vivo. The disclosed engineering strategy serves as a prominent platform for the design of other multifunctional materials for various biomedical challenges.

[1] Division of Engineering in Medicine, Department of Medicine, Brigham and Women's Hospital, Harvard Medical School, Boston, MA, USA. [2] Division of Health Sciences and Technology, Harvard University – Massachusetts Institute of Technology, Cambridge, MA, USA. [3] Nanomedicine Laboratory, Department of Chemical Engineering, Northeastern University, Boston, MA, USA. [4] Institute of Chemistry, São Paulo State University (UNESP), Araraquara, São Paulo, Brazil. [5] Multidisciplinary Center for Biological Research, University of Campinas (UNICAMP), Campinas, São Paulo, Brazil. [6] Brasil University, Itaquera, São Paulo, Brazil. [7] LIMAV - Interdisciplinary Laboratory for Advanced Materials, Materials Science & Engineering Graduate Program, UFPI - Federal University of Piaui, Teresina, Piaui, Brazil. [8] Department of Physics, Federal University of Piaui, Teresina, Piaui, Brazil. [9] Department of Chemistry, Massachusetts Institute of Technology, MIT, Cambridge, MA, USA. ✉email: samsonafewerki20@gmail.com; lobo@ufpi.edu.br

The design and preparation of fibrous biomaterial scaffolds for various tissue engineering and biomedical applications that simultaneously possess appropriate mechanical, biological, and hydrophilic properties remain a clinical challenge[1]. The development of biocompatible and biodegradable surgical biomaterials, with an ability to integrate into the host, is vital to promote healing and functional regeneration of damaged tissues or organs and protect the host from impairment or potential infections[2,3]. Nevertheless, at the same time, promotion of various cellular activities (such as cell attachment, proliferation, and differentiation) is a critical objective. To date, we have witnessed a plethora of new biomaterials with a wide range of properties suitable for various biomedical applications; however, the ideal biomaterial fulfilling all the aforementioned requirements is still undiscovered[4]. Nature has always been an inspiration for improved biomaterial design and the ultimate paragon with its highly sophisticated materials and chemicals developed through innovative and highly efficient strategies[5]. In this context, an approach to mimic nature in the design of an ultimate biomaterial could be through the flawless reproduction of nature's own biomaterials and corresponding extracellular matrices (ECM)[6]. In this pursuit, the selection of appropriate polymeric materials is eminent.

Here, polycaprolactone (PCL) is one of the most extensively employed synthetic biomaterials in tissue engineering and biomedical applications[7,8] due to its favorable properties such as biodegradability, biocompatibility, high permeability[9], good stability, and mechanical properties, and also its versatility and relatively low-price. However, its highly hydrophobic nature impedes favorable cell attachment and its slow degradation occasionally limits its applications. Hydrophilicity is important for allowing the infiltration of cells within the material and the transportation of water, nutrients, and waste to and from cells, to promote cell attachment, as well as endorse cell proliferation and differentiation[10]. Nevertheless, several strategies have been demonstrated for improving the hydrophilicity of PCL, such as blending it with hydrophilic polymers[11,12], surface functionalization with ECM components (e.g., proteins[8]) or plasma treatment (PT)[13,14]. PT is a versatile, quick, easy, and facile technology for the surface modification of various materials[15,16]. It has previously been demonstrated that PT of PCL nanofibers promotes biological properties due to enhanced hydrophilicity[13,17,18], moreover, various other strategies for improving cell behavior and interactions of PCL-based fibers have been demonstrated, such as the use of cold atmospheric plasma allowing for facile immobilization of gelatin on PCL nanofibers[19], double plasma treatment comprising liquid plasma treatment, and dielectric barrier discharge plasma modification[20]. However, PT can also reduce the mechanical strength of PCL[13].

Furthermore, employing only a PT approach for changing the hydrophobicity of a surface can eventually lead to recovery of the biomaterials to their hydrophobic nature due to the rearrangement of polymer chains[21]. Nevertheless, the devised integrated strategy disclosed in this work represents a broader and versatile concept than previous reports, and would allow for the possibility to functionalize various surfaces with a wide range of molecules, polymers, and functional groups adaptable to an extensive variety of properties and applications[22]. However, to overcome some of the limitations with PT mentioned above and to provide a solid fabrication process, PT can be merged with direct surface modification (DSM) leading to more permanent modification[22]. Nevertheless, additional challenges still endure in integrating optimal biomaterials with a facile, scalable, and cheap fabrication technology[23]. Electrospinning technology (ES) embodies a facile, low cost, and efficient strategy for the preparation of fibers of controlled structure and dimensions imitating the natural ECM[24].

To meet all of the aforementioned challenges, we envision a unique integrated strategy for designing and engineering innovative PCL-based materials. We intend to engineer PCL fibers through an integrated ES-PT-DSM strategy providing PCL methacrylated fibers (PCLMA). Herein, in the pursuit of improving the biological performance of the PCLMA, due to the inherent missing of an RGD (arginine-glycine-aspartic acid) epitope and/or other cell adhesion moieties, we aim to blend PCLMA with gelatin (denatured collagen) methacryloyl (GelMA) providing a unique fiber blend (PCLMA:GelMA)[25]. Gelatin is one of the most commonly employed natural biomaterials in biomedical research due to its superior biological performance and resemblance to the native ECM. However, limitations (such as poor mechanical properties and fast enzymatic degradation) of gelatin have to be considered[26]. Nevertheless, through this merging of biomaterials, limitations of gelatin can be overcome[27].

As a clinical application, we challenged the above-mentioned materials to treat hernias. A hernia is when organ tissue inserts through an opening in the muscle or tissue due to a loss of tissue structure or function[28]. Hernia repair is among the most common surgical procedures and remains a challenging clinical problem (with approximately 20 million performed annually), and global hernia repair devices and their associated consumable market are estimated to reach $6.1 billion by 2020[29]. A hernia can ensue as a result of a congenital defect, traumatic injury, or failed closure of a surgical wound (e.g., in the abdominal wall)[30]. An incarcerated hernia can in the worst case lead to infection, sepsis, and possible death of some parts of the organ or tissue if not operated on in time[31]. A surgical treated hernia normally ends up with the use of implants to promote the repair and provide support to the weakened tissue. Human abdominal tissue is comprised of several individual layers with different mechanical properties, such as the linea alba [uniaxial test method: elastic modulus ~70 kPa (transverse tensile stress) and ~8 kPa (longitudinal tensile stress)], the anterior [uniaxial: ultimate tensile stress ~8.3 MPa (transverse) and ~3.4 MPa (longitudinal)] and the posterior rectus sheath [uniaxial: ultimate tensile stress ~5.3 MPa (transverse) and ~2.0 MPa (longitudinal)][32]. However, the mechanical properties reported on the composite layer through non-invasive IR imaging/3D imaging have shown an elastic modulus of ~42.5 kPa (transverse) and ~22.5 kPa (longitudinal). Therefore, meeting these required physiological properties are crucial in order to engineer a suitable material for hernia repair[33].

In this context, various type of implants have been employed such as using biological prostheses (derived from humans or consisting of collagen or acellular elastin), porcine and bovine products or synthetic materials (absorbable, non-absorbable, or a combination as well as textile based and polypropylene, polyethylene, polytetrafluoroethylene, polyglactin 910, poliglecaprone 25, poly (L-lactide-co-glycolide), and terephthalate polyester based meshes)[30]. Despite the vast number of available implants for hernia repair, no ideal material for every surgery exists, and, furthermore, challenges with postsurgical complications such as infection, fibrosis, mesh rejection, and hernia recurrence still remain critical clinical challenges[34]. Motivated by this huge problem, challenge and need, we envision devising a biomaterial overcoming these challenges through our proposed integrated ES-PT-DSM strategy, thus, engineering an innovative nanofiber for hernia repair.

An ideal material for hernia repair applications should be chemically inert, not prompt any chronic inflammatory reaction, easy to handle, biocompatible, mechanically resistant, inexpensive and able to promote tissue ingrowth and integration of host tissue as well as neovascularization[30]. In this context, electrospun nanofibers could function as good candidates for hernia repair

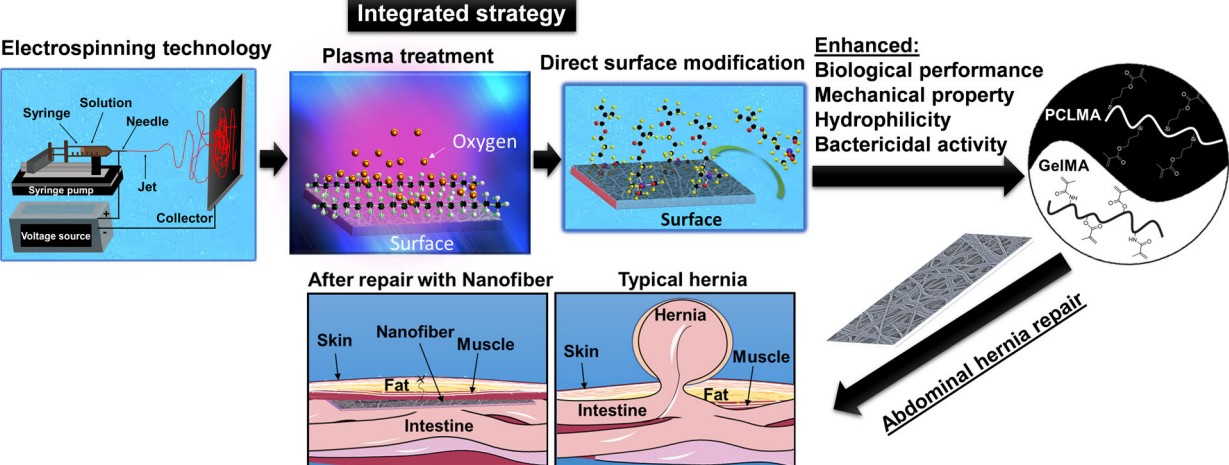

**Fig. 1 An overview of the integrated engineering strategy.** The approach comprises an integrated strategy for the design of an innovative polycaprolactone (PCL)-based nanofiber by combining a facile and low-cost electrospinning technology (ES) with plasma treatment (PT) and direct surface modification (DSM) (ES-PL-DSM) approach providing PCL methacrylate (PCLMA) nanofibers with improved properties (such as hydrophilicity, mechanical properties, bactericidal property, and biological performance). The biological performance can be further improved through the incorporation of gelatin methacryloyl (GelMA). The engineered nanofibers could function as a good candidate for abdominal hernia repair.

due to their mimicry of the three-dimensional (3D) structure of the ECM, ability to promote various cellular mechanisms, ultra-light weight, non-woven structure composed of nano-scale polymer fibers which can direct cellular orientation, and attenuation or disinclination towards producing an inflammatory response. Additionally, they are simple and cost effective to produce[35]. We have recently demonstrated the bactericidal property of electrospun nanofiber blends composed of a combination of PCL and polyethylene glycol (PEG):GelMA[12,36]. However, one challenge with the employment of electrospun nanofibers for hernia repair could be a lack of sufficient mechanical properties[31]. De facto, PCL electrospun fibers[37,38] or PCL based meshes[39] have previously been demonstrated as an optional candidate material for hernia repair.

Herein, we disclose an integrated strategy which comprises a combination of ES, PT, and DSM for the engineering of multifunctional fiber blends of PCLMA and GelMA for abdominal hernia repair. The devised nanofibers displayed bactericidal properties with tunable mechanical properties and with enhanced hydrophilicity and biological performance. The nanofibers successfully prompted the healing of in vivo hernia repair, which showed good biointegration, blood vessel formation, and tissue ingrowth.

## Results and discussion

**The engineering of multifunctional nanofibers through an integrated strategy.** An overview of the overall strategy of the integrated ES-PT-DMS approach used here and the combination of PCLMA and GelMA and its further application is depicted in Fig. 1.

The detailed steps for the devised integrated strategy are presented in Fig. 2a. Initially, the PCL fibers were obtained after an ES process and subsequently they were activated through PT introducing highly reactive oxygen-based groups (PCL-OH). Afterwards the activated fibers underwent DSM with 3-(Trimethoxysilyl)propyl methacrylate, providing the methacrylated PCL fibers (PCLMA) (Fig. 2a). The incorporation of silane allowed for further chemical modification of the versatile allyl moiety group or further photocrosslinking[22]. The success of the silane attachment was confirmed by proton nuclear magnetic resonance ($^1$H-NMR) analysis and the content of physically and

covalently bonded silane was also identified (Fig. 2b). In order to determine the quantity of covalently bonding to the modified nanofibers (PCLMA), the nanofibers underwent a reprecipitation and washing step ensuring the removal of all non-covalent bonded fragments (Supplementary Fig. 1). The final $^1$H-NMR analysis confirmed that the PCLMA contained 61 mol% of the silane, whereas 58% of the functionalization was physically bonded and 3% was covalently bonded. Prominently, if one would like to direct the reaction towards a higher degree of covalent bonding, we have previously reported an acid catalyzed silylation technology for the facile surface modification of various surfaces[22]. Furthermore, the developed PCLMA was combined with GelMA and electrospinning provided a PCLMA:GelMA electrospun blend (Fig. 2c).

Ensuring well integrated and mechanically robust nanofiber blends, we performed a dual crosslinking strategy. The first crosslinking step was performed to crosslink the gelatin moiety with the aid of glutaraldehyde (Fig. 2d). This step is vital since gelatin is the moiety weakening the blended fiber due to its immediate disintegration upon exposure to an aqueous environment. Subsequently, the second crosslinking supports a good integration between the PCLMA and GelMA through crosslinking of the methacrylate groups on and in between both components. This step introduces a second covalent bonding in the system (Fig. 2d). The success of each crosslinking step was confirmed through Fourier-transform infrared spectroscopy (FTIR) analysis. Post the glutaraldehyde step, the disappearance of the primary amine peaks at a wavenumber around 3630–3730 cm$^{-1}$ confirmed the crosslinking between the amines in GelMA (Fig. 2e, i). Moreover, the UV-crosslinking steps were confirmed by fading of the bands at 1634 and 810 cm$^{-1}$ corresponding to the carbon–carbon double bonds (C=C) (Fig. 2e, i). The success of the crosslinking of the PCLMA and the nanofiber blend (PCLMA:GelMA) was also confirmed by the absence of the peaks corresponding to the C=C bonds (Fig. 2e, i–iii).

**Optimization and mechanical characterization.** Since each of the components (PCLMA and GelMA) contribute to its inimitable properties (mechanical, wettability, cell compatibility, etc.), having an optimal ratio of such components is critical for obtaining a superior blend exhibiting strong mechanical

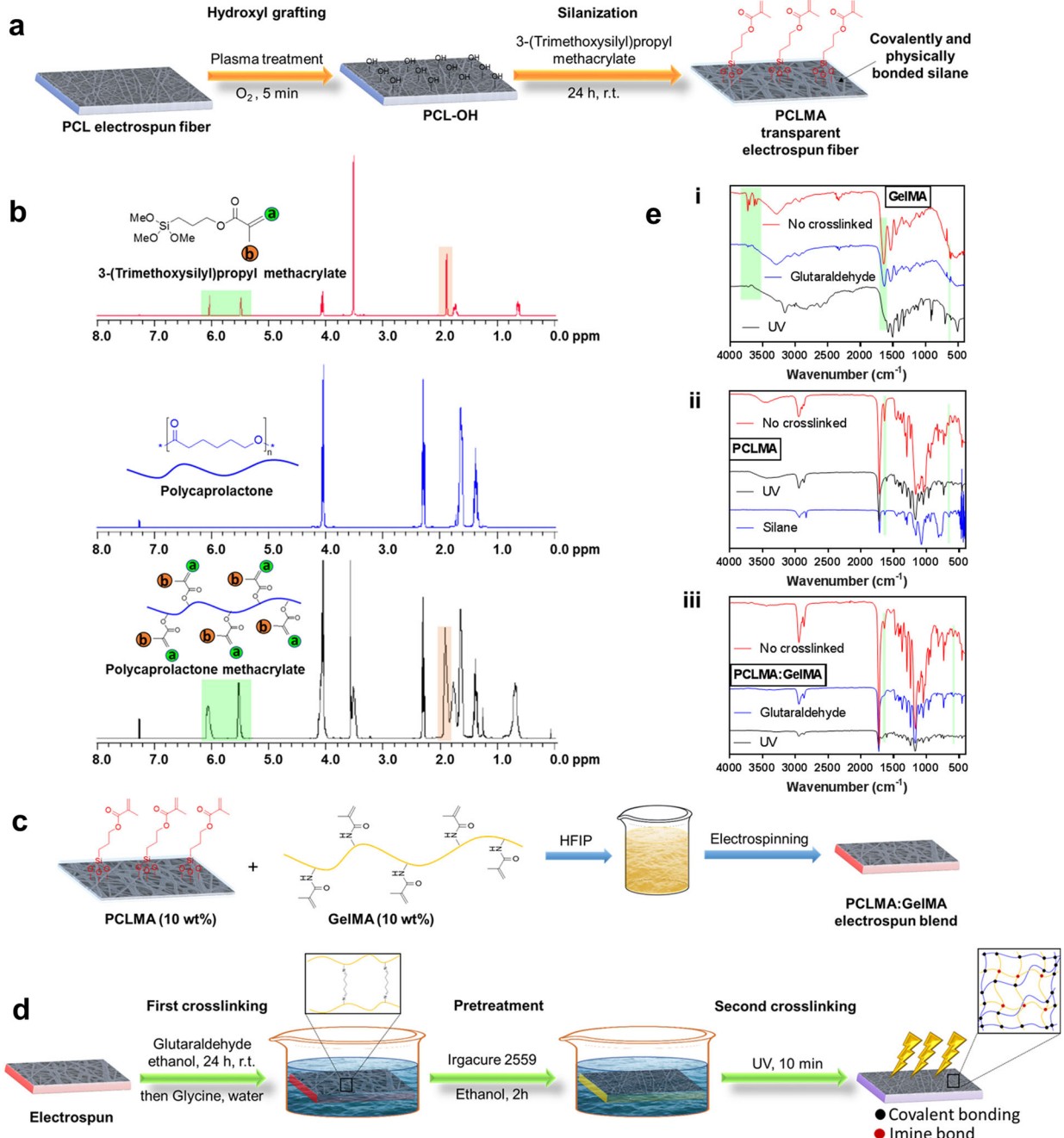

**Fig. 2 The fabrication and characterization of the engineered fibers. a** Fabrication of the polycaprolactone (PCL) electrospun fiber. Oxygen plasma treatment (PT) for the introduction of reactive hydroxyl groups followed by a direct silanization treatment providing physically and covalently bonded silane methacrylate onto the PCL fiber (PCLMA). **b** The $^1$H NMR of 3-(Trimethoxysilyl)propyl methacrylate, PCL and PCLMA. **c** The preparation of PCLMA and GelMA blend electrospun fibers (PCLMA:GelMA) in Hexafluoroisopropan-2-ol (HFIP). **d** The crosslinking of the designed electrospun fibers. The first crosslinking step proceeds through glutaraldehyde treatment, followed by glycine quenching and then the second UV-crosslinking step provides the final crosslinked product. **e** The Fourier-transform infrared spectroscopy (FTIR) spectra of the designed nanofibers prior and post each fabrication step confirming the success of the first and second crosslinking steps for **i**, GelMA, **ii**, PCLMA and **iii**, PCLMA:GelMA.

properties and at the same time good biological performance. In order to fine tune the composition, a wide range of various fibers of different ratios were prepared and primarily their mechanical properties were first investigated through tensile experiments using a mechanical tester (Fig. 3a, b). As expected, fibers containing solely gelatin displayed weak mechanical strength (0.0313 MPa), however, the GelMA fiber displayed a ~4-fold increase (0.115 MPa) (Fig. 3a, entries 1 and 2). The PCL showed a tensile strength of 1.23 MPa and the PCLMA exhibited a 4-fold decrease

in tensile strength (0.293 MPa) compared to the PCL fiber (Fig. 3a, entries 3 and 4). Next, we mixed various ratios of PCLMA:GelMA (90:10–10:90) in order to identify the optimal ratio. Eventually, the composition of PCLMA:GelMA equaling 70:30 displayed the highest mechanical properties with tensile strengths of 0.386 MPa (Fig. 3a, entry 9).

Simultaneously, the produced fibers are intended for biomedical applications and, thus, their stability in aqueous media is important. Therefore, the stability of the nanofibers in aqueous

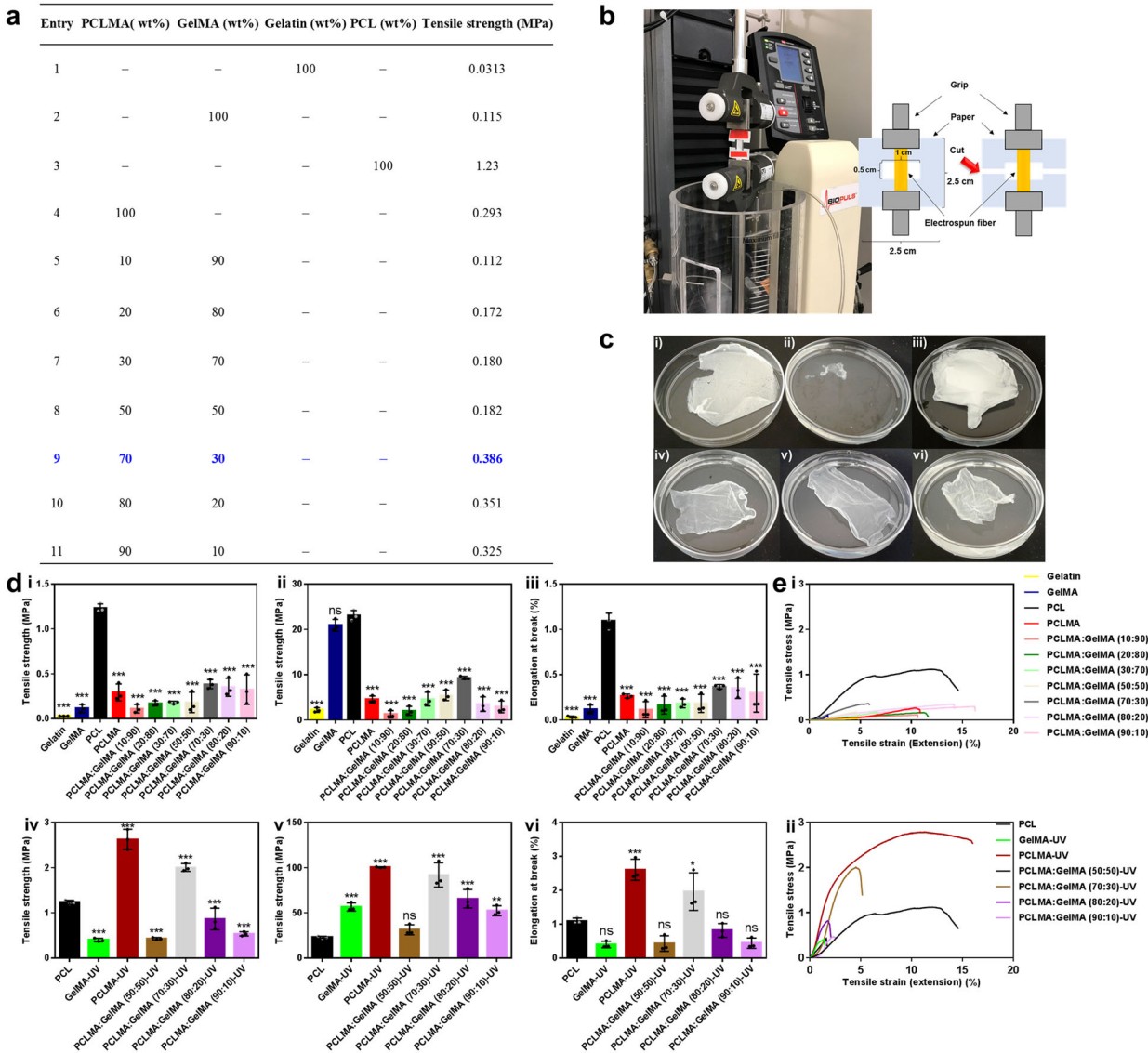

**Fig. 3 Screening studies and mechanical evaluation of various nanofiber compositions. a** Table demonstrating the various compositions and their respective tensile strength. All the pure fibers were prepared in a 10 wt.% stock solution and mixed in various ratios, therefore, the wt.% in the table indicates wt.% employed of the stock solution. For all the sample mixtures prior to electrospinning, Hexafluoroisopropan-2-ol (HFIP) was employed as the solvent and the final volume for each solution was 5 ml. **b** The tensile test setup of the electrospun fibers with a mechanical tester. **c** Images of the electrospun fibers. i GelMA fiber in a petri dish. ii GelMA fiber in phosphate-buffered saline (PBS) immediately disintegrated. iii The hydrophobic and stable PCL fiber in PBS solution. iv The stable PCLMA fiber in PBS. v The PCLMA:GelMA (70:30) fiber blend in PBS. vi The PCLMA:GelMA (70:30) fiber blend after UV crosslinking in PBS. **d** Mechanical data of the various electrospun fibers before crosslinking i tensile strength, ii elastic modulus, and iii elastic elongation at break, and correspondingly after crosslinking iv tensile strength, v elastic modulus and vi and elastic elongation at break. **e** The tensile stress vs. tensile strain data for the samples before and after crosslinking. Values are mean ± SD, N = 3. ANOVA (p < 0.05) following by Tukey's multiple comparisons test. The data were compared to solely PCL (*) p < 0.05, (**) p < 0.01, (***) p < 0.005, ns: not significant, mean statistical differences. The numeral post the fiber name indicates the ratio of the materials of the respective blend (e.g., PCLMA:GelMA (10:90) means 10% PCLMA and 90% GelMA).

media was studied, and, as expected, the fibers containing solely GelMA were not stable in an aqueous solution, and consequently, immediately dissolved when placed in a phosphate-buffered saline (PBS) solution (Fig. 3c, i–ii and Supplementary Movie 1). The PCL fibers in the water solution floated due to their hydrophobic nature (Fig. 3c, iii). Interestingly, the PCLMA fibers were transparent and did not float on the surface compared to the pure PCL fibers (Fig. 3c, iv); correspondingly, the PCLMA:GelMA at the 70:30 fiber blend ratio showed the best stability both prior and post crosslinking (Fig. 3c, v–vi). PCL fibers prior to crosslinking, as expected, showed the highest tensile strength

(1.23 MPa), elastic modulus (23.1 MPa) and elongation at break (1.09 %) (Fig. 3a, d, i–iii). However, of the various blends, the PCLMA:GelMA at the 70:30 blend ratio exhibited the highest tensile strength (0.386 MPa), elastic modulus (9.29 MPa) and elongation at break (0.366%) (Fig. 3a, d, i–iii). Due to the instability of the GelMA fibers in aqueous solution, only the fiber blends containing at least 50% PCLMA were stable enough for mechanical testing. The remaining either disintegrated, dissolved or their nanofiber structure was destroyed during the crosslinking procedure. Nevertheless, prominently after crosslinking, the PCLMA fibers improved for all features compared to pure PCL.

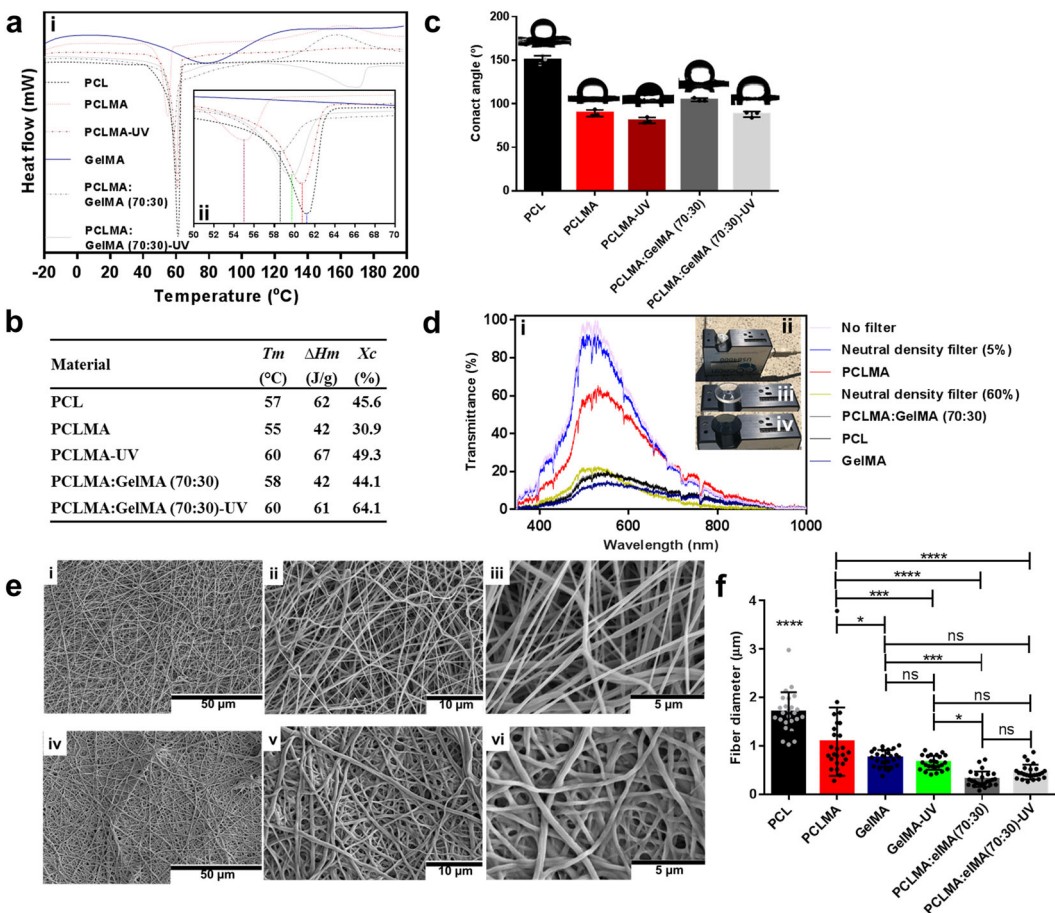

**Fig. 4 Thermal, hydrophilicity, transparency, and morphology characterizations. a**, i Differential scanning calorimetry (DSC) thermograms of the various fibers. ii Zoomed area. **b** The thermal analysis of the fibers providing the $T_m$ (crystalline melting temperature), $\Delta H_m$ (melting enthalpy) and $X_c$ (fiber crystallinity). **c** The contact angles measured and the images of the water drops on the fibers captured during the analysis. Values are mean ± SD, $N = 3$. **d**, i Light transmittance property analysis of the various nanofibers. ii Demonstrating the setup applying the transparent PCLMA fiber, iii neutral density filter 5% and iv neutral density filter 60% of absorbance. **e** Characterization of the morphology of the electrospun nanofibers through scanning electron microscopy (SEM) analysis with various magnifications of i–iii, PCLMA:GelMA (70:30) and the iv–v PCLMA:GelMA (70:30)-UV. **f** The fiber diameter for PCL, PCLMA, GelMA, GelMA-UV, PCLMA:GelMA (70:30), and PCLMA:GelMA (70:30)-UV. Values are mean ± SD, $N = 25$. ANOVA ($p < 0.05$) followed by a Tukey's multiple comparisons test. *$p < 0.05$, ***$p < 0.005$, ****$p < 0.001$, ns not significant, mean statistical differences.

The tensile strength increased to 2.63 MPa and the elasticity of the fibers correspondingly increased with an elastic modulus of 100.5 MPa and elongation at break at 2.61% (Fig. 3d, iv–vi). The optimal blend (PCLMA:GelMA (70:30)) also showed improved mechanical properties (tensile strength (2.00 MPa), elastic modulus (92.0 MPa), and elongation at break (1.96%)). These optimal mechanical properties were also confirmed by comparing the tensile stress vs. tensile strain in graphs (Fig. 3e, i–ii).

**Thermal, hydrophilicity, and transparency characterization.** Furthermore, the thermal stability of the nanofibers was evaluated through differential scanning calorimetry (DSC) analysis providing the thermal parameters, crystallization temperature ($T_c$), melting temperature ($T_m$), enthalpy of fusion ($\Delta H_m$), and degree of crystallinity ($X_c$) (Fig. 4a, b)[40]. The crystallization behavior of the PCLMA fiber slightly shifted towards a lower temperature compared to the PCL fibers. The $X_c$ and $\Delta H_m$ decreased from 45.6 to 30.9% and 62 to 42 J/g, respectively, whilst the $T_m$ slightly decreased from 57 to 55 °C (Fig. 4a, b). After incorporation of GelMA to the nanofibers, the $X_c$ increased to 44.1% and the $T_m$ to 58 °C, whilst the $\Delta H_m$ remained the same as the PCLMA. However, to our delight, the thermal properties of all the samples

increased post UV crosslinking, the PCLMA fibers increased the $X_c$ to 49.3%, $\Delta H_m$ 67 J/g and the $T_m$ to 60 °C and for the fiber blend (PCLMA:GelMA(70:30)) as well to 64.1%, 61 J/g, and 60 °C, respectively (Fig. 4a, b).

Furthermore, the wettability and hydrophilicity of the fibers were evaluated through contact angle (CA) and surface energy measurements[41]. The superhydrophobic PCL fibers displayed a CA of 151°, but the PCLMA showed a more hydrophilic nature with a CA of 89°, which decreased even further post UV crosslinking (CA = 81°) (Fig. 4c). The nanofiber blend (PCLMA: GelMA (70:30)) showed a CA of 105°, but after crosslinking, the hydrophilicity increased to a CA of 88°. Generally, higher surface energy corresponds to lower CA[42], and similar trends were observed for the devised nanofibers. The surface energy of the PCLMA post UV-crosslinking was 38.3 mJ/m² and for the blended nanofiber 46.6 mJ/m² prior to crosslinking and 44.3 mJ/m² after crosslinking, respectively (Supplementary Table 1).

Surface energy is an important parameter playing a significant role in regulating bacteria adhesion to surfaces[43]. Interestingly, it has been previously stated that a surface energy of 42.5 mJ/m² is optimal to reduce bacteria adhesion and colonization, which supports our findings[44]. Furthermore, the transparency of the nanofibers was characterized through transmittance measurements

using sun light as the light source and neutral density filters (5 and 60% of absorbance) as the reference (Fig. 4d). Remarkably, the transparent PCLMA nanofibers displayed a transparency of up to 63% for visible light. Transparent films are highly desirable and important in various applications. For instance, transparent wound dressings allow for the continued visualization of the wound, avoiding the removal of the dressing which could potentially lead to contamination and disruption of healing[45].

**Morphology of the nanofibers**. The morphology of the electro-spun fibers was further characterized through scanning electron microscopy (SEM) analysis which displayed smooth nanosized and microsized fibers in the absence of defects or beads on the scaffolds (Fig. 4e and Supplementary Fig. 2). The analysis further confirmed that the PCL fibers displayed the largest diameter structures with a diameter of $1.7 \pm 0.41$ μm, followed by the PCLMA at $1.1 \pm 0.70$ μm. The rest of the designed electrospun fibers GelMA ($0.75 \pm 0.17$ μm), GelMA-UV ($0.65 \pm 0.14$ μm), PCLMA:GelMA (70:30) ($0.32 \pm 0.16$ μm), and PCLMA:GelMA (70:30)-UV ($0.46 \pm 0.15$ μm) showed nanosized structures (Fig. 4f). It is worth highlighting that our main material groups (PCLMA:GelMA (70:30) and PCLMA:GelMA (70:30)-UV) did not show noteworthy changes in morphology after crosslinking, where smooth, homogenous and non-defected surfaces were obtained.

**In vitro stability and viability studies**. Encouraged by these findings, we proceeded to investigate the in vitro stability per-formance of the nanofibers. The enzyme collagenase is known to degrade gelatin in vivo, and lipase has been shown to similarly wear out PCL. Thus, we performed an in vitro degradation study with collagenase Type II (1.0 μg/mL) and lipase Type VII (1.0 μg/mL), respectively, at physiological conditions. Interestingly, PCLMA and PCLMA-UV demonstrated high stability when exposed to the lipase condition and did not show any degrada-tion, whilst the PCL showed a slow degradation with 50% of mass loss after 65 days (Fig. 5a, i). Moreover, the nanofiber blends (PCLMA:GelMA (70:30) and PCLMA:GelMA (70:30)-UV) dis-played a slow and sustained degradation in both of the exposure conditions investigated. The lipase system was linked to the fastest rate of degradation of the blended fibers, where 72% of the PCLMA:GelMA (70:30) and 66% of the PCLMA:GelMA (70:30)-UV fibers degraded after 65 days (Fig. 5a). Considering translational applications, the fibers could be sterilized by various established and known procedures such as β-irradiation, UV-irradiation, or by the use of ethylene oxide[46,47]. Furthermore, the fibers were evaluated for their biocompatibility in terms of NIH/3T3 fibroblast cell adhesion, viability, morphology, and pro-liferation. Cells were cultured on each of the fibers PCL, gelatin, GelMA, GelMA-UV, PCLMA, PCLMA-UV, PCLMA:GelMA (70:30), and PCLMA:GelMA (70:30)-UV fiber membranes with an initial cell seeding density at $0.05 \times 10^6$ cells per fibers. The cell viability was determined using calcein AM and ethidium homo-dimer I (EthD-1) at day 7, following the manufacturer's instructions. Similarly, cell morphology and cell proliferation were observed by F-actin staining of the cells grown on the membranes, using Alexa Fluor® 488-phalloidin at day 7. The cells were counterstained with DAPI for nuclei and imaged using fluorescence microscopy. The results have shown that, except for the PCL nanofibers surface, none of the other membranes showed any toxic effects to fibroblasts, and that they all supported cell attachment and growth. No cell attachment was observed to the PCL nanofibers due to its highly hydrophobic nature[12].

The viability of the cells was found to be more than 95% for all of the membranes tested except for the PCL membrane (Fig. 5b).

Fluorescence imaging of F-actin/nuclei-stained cells indicated green staining for F-actin in the cytoplasm while the blue stains for nuclei (Fig. 5c). The cells were homogenously distributed within the respective nanofiber membranes, exhibiting high viability and proliferation of cells on the membranes except for the PCL nanofiber membrane. Thus, the engineered nanofibers support cell adhesion, growth, and proliferation without requiring a surface coating with bioactive molecules.

**In vivo biocompatibility study**. The biocompatibility and tissue integration of the engineered nanofibers were further evaluated in vivo through subcutaneous implants for 5 days in the dorsal region of rats. The histological results were evaluated employing the histological grading scale for soft tissue as presented in Sup-plementary Table 2. Primarily, the study confirmed the normal appearance of the adjacent tissue and absence of any necrosis for all the groups analyzed (Fig. 5d, e). Moreover, the capsules observed surrounding the implants did not display any significant difference among the nanofibers (Fig. 5d, i). A significantly greater degree of tissue response of the capsule surrounding the implants was observed around the crosslinked PCLMA:GelMA (70:30) implant compared to the PCL and GelMA samples (Fig. 5d, ii). Similarly, PCLMA:GelMA (70:30), PCLMA and GelMA demonstrated a tissue response directly adjacent to the implant surface (interface) which was significantly higher as compared to PCL (Fig. 5d, iii). Additionally, the PCLMA:GelMA (70:30) showed a significant increase in the number of blood vessels compared to PCL and GelMA (Fig. 5d, iv). These results could also be visualized in the presented photomicrographs (Fig. 5e).

**Abdominal hernia repair application**. The engineered nanofi-bers were further investigated in vivo as a potential scaffold candidate in ventral abdominal hernia repair. The ventral abdominal hernia repair mouse model was performed on the B6/cba F1 mice strain (Fig. 6a)[48]. All of the various PCL based nanofibers (PCL, PCLMA, PCLMA-UV, PCLMA:GelMA (70:30), and PCLMA:GelMA (70:30)-UV) applied to the hernia lesion showed good interactions with the abdominal muscle tissue (Fig. 6b–d). The tissue analysis demonstrated inflammatory pro-cesses as shown by inflammatory cells around the materials due to their natural mechanisms of decomposition. Despite the good compatibility of all the materials, some of them were better than others. For instance, the PCLMA:GelMA (70:30) group proved to be less inflammatory than the other groups and had a thinner layer of inflammatory cells, what may lead to a better regenerative pathway. On the other hand, the PCL and PCLMA displayed a massive presence of inflammatory cells with many surrounding fibroblasts. Moreover, giant phagocytic cells and macrophages at the interface of the membrane and tissue were much higher with the PCLMA:GelMA (70:30), indicating a more conspicuous bio-degradation and a cleaning tissue process that facilitates regen-eration activity. Focusing on the edge of the interaction, we observed a good intersection between muscle tissue and healing connective tissue, showing good vascularization and a healthy appearance. Despite that all the samples demonstrated collagen tissue formation, the PCLMA showed surrounding collagen tissue that was more immature and while the PCLMA:GelMA (70:30) showed mature collagen formation and a greater density of col-lagen that, like the PCLMA, had a higher number of stretch fibers present. This is expected considering that this tissue is under tensile strength and thus the formation of these stretch fibers; as stress fibers (composed of actin and non-muscle myosin II) as well as elastic fibers, are valuable for new tissue recovery. More-over, the crosslinked fibers (PCLMA-UV and PCLMA:GelMA

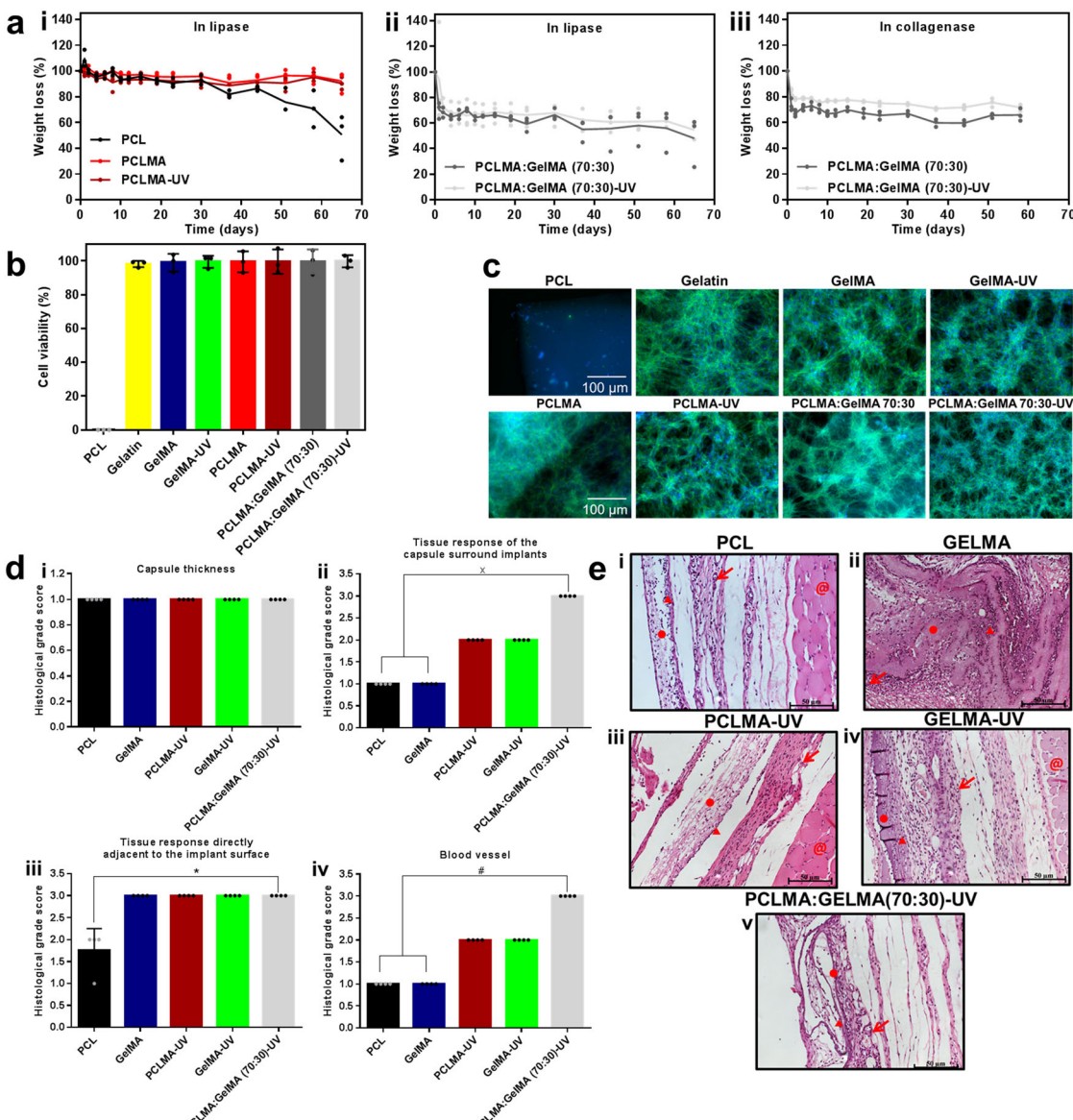

**Fig. 5 In vitro stability and viability evaluation and in vivo compatibility studies. a** In vitro degradation study of the engineered fiber scaffold. **i** PCL, PCLMA and PCLMA-UV in a lipase system, PCLMA:GelMA (70:30) and PCLMA:GelMA (70:30)-UV in ii, lipase and in iii, collagenase Type II systems, respectively. Values are mean ± SD, $N = 3$. **b** Cell viability experiments on the various fibers using NIH/3T3 fibroblast cells after 7 days. Values are mean ± SD, $N = 3$. **c** Proliferation and spreading images of the fibers on NIH/3T3 fibroblast cells after 7 days. Histological evaluation after subcutaneous implantation in the dorsal region of rats for five days. **d**, i Capsule thickness ii Tissue response of the capsule surround implants iii Tissue response directly adjacent to the implant surface and **iv** blood vessel of the subcutaneous implants using the histological grading scale (Supplementary Table 2). Error bars represents mean ± SD of the mean, N = 3; × $p < 0.05$ compared with PCL and GelMA; *$p < 0.05$ compared with PCL; #$p < 0.05$ compared with PCL and GelMA. **e** The photomicrographs of subcutaneous implants of i PCL, ii GelMA, iii PCLMA-UV, iv GelMA-UV and v PCLMA:GelMA (70:30)-UV. ●Nanofiber, ←Blood vessel and ▲cell inflammatory and @panniculus carnosus muscle. Hematoxylin and Eosin staining (H & E staining), scale bar = 50 μm.

(70:30)-UV) were linked to a decrease in the formation of stretch fibers, and a bias to produce less collagen. This outcome could be due to the higher stiffness of the fibers. The uncrosslinked fibers were more flexible and more adaptable to fit on the lesion spot. On the other hand, crosslinking seemed to favor blood vessel formation. Nevertheless, except for the PCL fibers, all of the other groups demonstrated a good number of new blood vessels formed.

Various PCL-based electrospun fibers for hernia applications have previously been presented[49,50], for instance, drug-loaded nanofibers with the antibiotic levofloxacin and antibacterial agent irgasan for the prevention of potential bacterial infections[38]. Nevertheless, this approach might promote the prevalence of multiantibiotic resistant organisms[51]. Moreover, PCL and polypropylene were combined for the generation of nanofibers with various orientations and patterns that demonstrated good mechanical properties and tunable cell morphology based on patterning[52]. However, in the presented technology, the PCL is combined with GelMA which is known to promote various cellular and biological activities and tissue repair[27]. As *vide supra* highlighted, the presented technology represents a wider and more versatile method compared to the previous studies, where it could allow for facile tailoring of the engineered materials, for instance, for the effortlessly addition of active ingredients or desired properties such as fluorophores, etc. due to the important functional groups presented during fabrication[22].

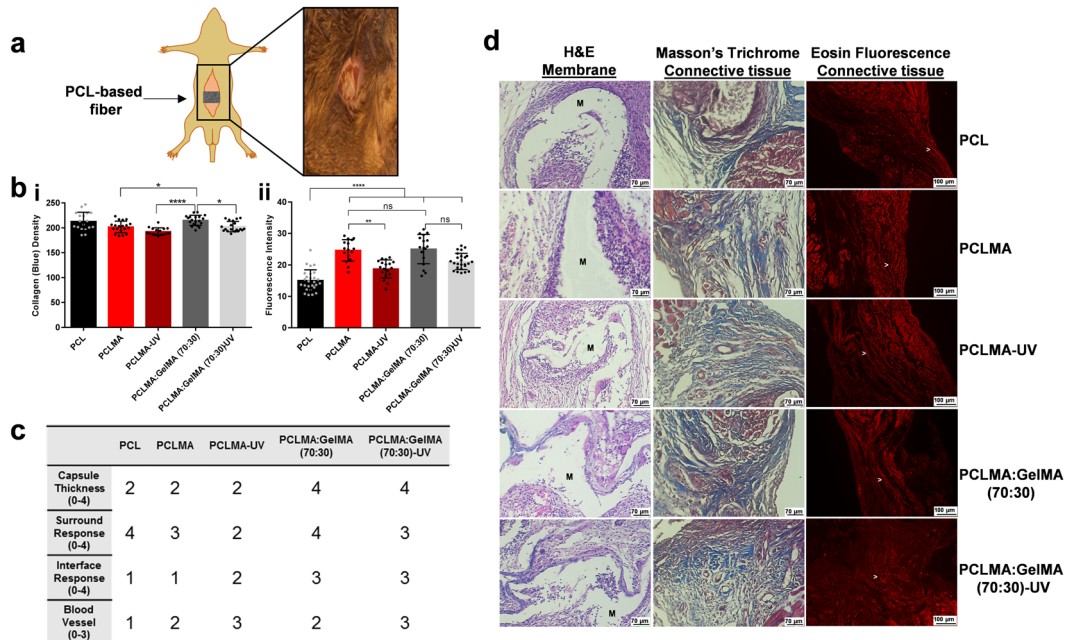

**Fig. 6 The in vivo performance of the engineered nanofibers in a ventral abdominal hernia repair mouse model. a** Schematic and picture demonstrating the strategy for the in vivo experiments of abdominal hernia repair. **b** Quantification of connective tissue formed as a result of the hernia injury recovery employing the various fibers after 29 days of implantation. i Total collagen density of the surrounding tissue injury by Masson's Trichrome analysis and processed by ImageJ software. ii Stretch fiber amount distinguished by eosin fluorescence under a 590–630 nm filter of surrounding tissue injury as processed on ImageJ software. Error bars represents mean ± SD of the mean, N = 3; *p ≤ 0.05, **p ≤ 0.01, ****p ≤ 0.0001, ns (no significance). **c** Scoring table showing subjective data regarding the recovery of the hernia injury treated with the various PCL-based nanofibers after 29 days of surgery recovery (higher number improve rating). **d** Histological analysis results after 29 days of implanting the nanofibers using Hematoxylin Eosin (H&E) showing the material interface tissue distinguishing the inflammatory cells and giant multinucleated cells promoting material degradation and Masson's Trichrome and Eosin Fluorescence which both distinguish connective tissue (blue and red respectively) on the surround tissue. Membrane (M = Fibers) and stretch fibers (white arrowhead).

Considering regulatory aspects, since hernia support scaffolds are generally considered as medical devices and not as drugs, this will have considerable milder regulatory process[53]. Furthermore, the engineered nanofibers in this work are comprised of PCL, which is an already United States Food and Drug Administration (FDA) approved material for certain applications[39], and according to the FDA if the new device is equivalent to previous device it will be safe and effect as the counterpart, nevertheless supporting safety data may be necessary[54]. Generally, bringing a medical device to the market can take about 3−7 years, whilst about 12 years for new drugs or drug/medical device combination products[55].

**The bactericidal effects of the engineered nanofibers.** Additionally, the bactericidal properties of the electrospun fibers are vital, in particular for hernia repair since infection is a main clinical challenge, thus, having a material with the ability to prevent any potential future infection would be of great benefit[34,51]. The bactericidal tests were performed against the three bacteria strains: gram-positive *Staphylococcus aureus* (*S. aureus*) (Fig. 7a, i), gram-negative *Pseudomonas aeruginosa* (*P. aeruginosa*) (Fig. 7a, ii) and the drug resistant *Methicillin-resistant Staphylococcus aureus* (MRSA) (Fig. 7a, iii) in order to evaluate the potential for the nanofibers to deter or inhibit bacteria adhesion and proliferation. The results from the various fibers were compared to the results from the pure PCL fibers. The various tests performed included colony counting experiments, fluorescence quantification of intracellular ROS, and electron microscopy investigation. From the direct quantification of bacterial colonies through conventional plating methods, significant differences in the anti-pathogenicity of the fibers were

determined. For all the microbes evaluated, the PCLMA fibers contributed an immediate appreciable and critical reduction in colonizing bacteria (Fig. 7a). After 23 h of exposure to *S. aureus*, *P. aeruginosa*, and MRSA, PCLMA fibers were colonized by 17×, 213,199×, and 27.2× fewer bacteria respectively relative to pure PCL fibers; these corresponded with 1.23-log, 5.33-log, and 1.44-log reductions. With the intrinsic consolidation of the blended nanofiber (PCLMA:GelMA (70:30)), bacterial adhesion was acute. However, significant diminutions in the numbers of bacteria colonizing the blended fiber following UV-crosslinking (PCLMA: GelMA (70:30)-UV) were observed, and this result is especially impactful for *P. aeruginosa*. Specifically, following 23 h of culture, 3.88×, 18.4×, and 1.49× fewer cells, corresponding to 0.59-log, 1.27-log, and 0.17-log reductions, were observed for *S. aureus*, *P. aeruginosa*, and MRSA, respectively, along the UV-crosslinked composites than on pure PCL fibers. In our previous study, we have shown that the nanofiber blend after UV-crosslinking demonstrated significant bacteria reduction[12,36]. Importantly, this was all accomplished without resorting to the use of antibiotics.

To better understand the mechanisms of the anti-bactericidal activity of the fibrous scaffolds and their specific interactions with the bacteria, we determined reactive oxygen species (ROS) generation inside the live bacteria and undertook SEM analysis. Results obtained from the ROS study supported the colony counting results with low bacteria numbers coupled with high intracellular ROS for the PCLMA and PCLMA:GelMA (70:30)-UV fibers (Fig. 7a, b). These observations indicated a lower cellular attachment to the scaffolds due to their improved bactericidal activity and therefore bacteria growth inhibition (Fig. 7b, i–iii). Specifically, relative to the PCL scaffolds, bacteria on PCLMA and PCLMA:GelMA-UV expressed similar ROS

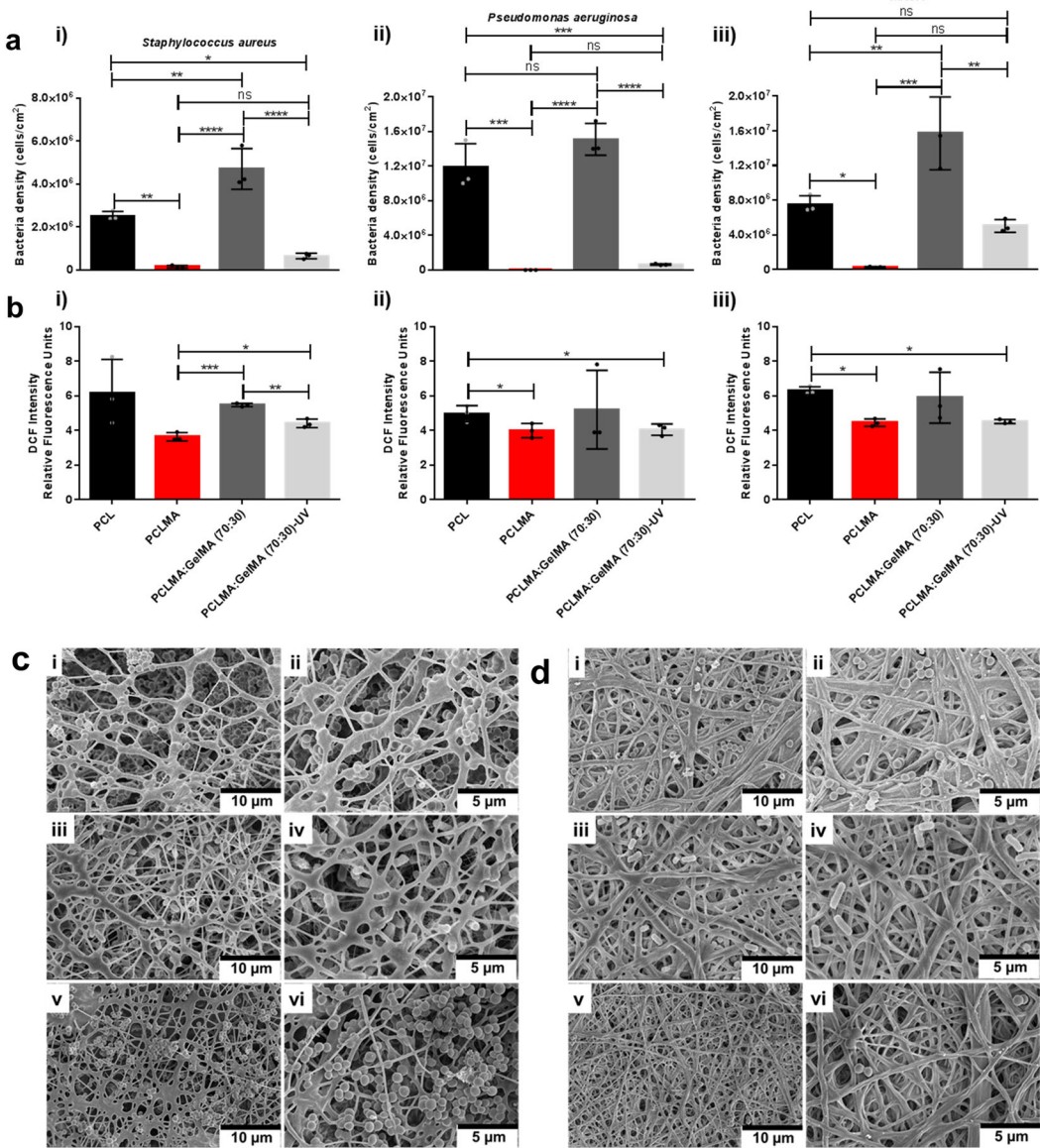

**Fig. 7 Bactericidal results on the various fibers (1 cm×1 cm) after 23 h. a** The colony-forming unit data for i *Staphylococcus aureus* (*S. aureus*), ii *Pseudomonas aeruginosa* (*P. aeruginosa*), and iii *Methicillin-resistant Staphylococcus aureus* (MRSA). **b** Reactive oxygen species (ROS) generated from fibers against i *S. aureus* ii *P. aeruginosa* and iii MRSA. Values are mean ± SD, *N* = 9. ANOVA (*p* < 0.05) following by Tukey's multiple comparisons test, *\*p* < 0.05, \*\**p* < 0.01, \*\*\**p* < 0.005 \*\*\*\**p* < 0.001, ns not significant, mean statistical differences. Scanning electron microscopy (SEM) images of the **c** PCLMA:GelMA (70:30) fibers pre-treated with the biofilm-forming i, ii *S. aureus*, iii, iv *P. aeruginosa*, and v, vi MRSA for 23 h, shown at high (scale bar 10 μm) and low magnifications (scale bar 5 μm). The SEM images of the **d** PCLMA:GelMA (70:30)-UV nanofibers pre-treated with the biofilm-forming i, ii *S. aureus* iii, iv *P. aeruginosa*, and v, vi MRSA for 23 h, shown at high (scale bar 10 μm) and low magnifications (scale bar 5 μm).

levels, except in the case of MRSA, where the levels of ROS generated increased significantly for PCLMA:GelMA-UV compared to PCLMA. An overall reduction in the relative fluorescence (RF) of bacteria in contact with the PCLMA and PCLMA:GelMA-UV fibers could be explained by a drop in the number of viable bacteria on those substrates. However, when RF values were adjusted to cell numbers obtained from the colony counting assays, for all the tested bacteria, an increase in the ratio of intracellular ROS per bacteria for both PCLMA and PCLMA: GelMA (70:30)-UV when compared with that obtained for the cells that had been cultured on PCL fibers, was observed. This effect was the most prominent for the Gram-negative bacteria assayed, *P. aeruginosa*.

To further examine the critical disparity in the bacterial concentrations colonizing the blended fibers both before and after

UV-crosslinking, SEM analysis was performed. From a visual analysis of randomized regions on the blended fibers, the aforementioned results were further confirmed pointing towards a reduction in the adherent bacteria on the UV irradiated composite fibers (Fig. 7c, d). Moreover, it could be clearly observed that the blended fibers prior to UV-crosslinking presented a fiber surface which was less dense with more pores allowing for bacteria to enter and promote biofilm-formation, whilst the nanofibers after crosslinking demonstrated less bacteria and biofilm-formation due to a more compact fiber scaffold.

## Conclusion

Herein, we disclose an integrated strategy (ES-PT-DSM) for the engineering of innovative multifunctional nanofibers with

amended properties and biological performance. The integrated strategy comprising a combination of ES, PT, and then further DSM (ES-PT-DSM) allowed for the facile fabrication of methacrylated PCL fibers (PCLMA). Moreover, through the combination with GelMA, the biological performance of the PCLMA fibers was further improved. For a scale-up process of the disclosed strategy, the fabrication methodology would most likely be employed in a one-pot or sequential manner, where each step of the fabrication process would be integrated into one reaction batch or several and each process are performed consecutively. Moreover, the silylation steps would most likely be performed in the presence of a catalyst shortening the time notably.

The mechanical properties of the blend could be tuned by varying the ratios of the two fibers and further crosslinking can enhance the mechanical strength. Furthermore, the initial in vitro cell viability study confirmed the high compatibility of the nanofibers. This was further corroborated through in vivo biocompatibility tests by subcutaneous implantation in the dorsal region of rats. The results demonstrated the absence of necrosis, normal behavior of the adjacent tissues, and a standard tissue response. Additionally, based on the in vivo hernia experiments and the analysis from the scoring on the histological data, it could be concluded that the PCLMA:GelMA (70:30) fibers demonstrated the highest scoring, and thus, showed less of an inflammatory response, good biodegradation, and collagen formation. Moreover, clear stress fibers and blood vessel formation were observed. Comparing the fibers with and without crosslinking, despite the improved mechanical properties of the fibers after crosslinking, they demonstrated slightly less formation of collagen and stretch fibers due to their higher stiffness. However, crosslinking is a parameter that can be tuned and adapted to the application, and thus, it is possible to adjust the optimal mechanical properties in order to promote the formation of collagen and the formation of blood vessels. Moreover, the engineered fiber scaffolds displayed bactericidal activity against the three common bacteria: S. aureus, P. aeruginosa, and MRSA. Particularly, the engineered PCLMA and the nanofiber blend, and prominently, the nanofiber after crosslinking (PCLMA:GelMA (70:30)-UV) showed a significant ability to reduce biofilm-formation and bacterial adhesion and to inhibit bacteria growth. These results were verified by various antibacterial tests (CFU and ROS analysis) and SEM analysis. Importantly, due to the bactericidal activity of the engineered nanofibers, they could prevent further potential infections during hernia repair without needing to resort to the use of antibiotics. To conclude, the PCLMA:GelMA (70:30) with and without crosslinking showed superior results compared to the other groups, and the blend is presented here as a good candidate for the treatment of abdominal hernia repair. Further investigation and studies of the ability of the nanofiber to prevent any future infection are ongoing in our laboratory and will be reported in the near future. We envisage the engineered nanofibers will find its way to the clinics and solve challenging biomedical problems besides in hernia applications, moreover, we hope the integrated strategy will pave the way for the engineering of other tailor-made multifunctional biomaterials with other strong potential candidates in various applications.

## Methods

**Materials**. The following chemicals were purchased from Sigma-Aldrich (St. Louis, MO, USA): PCL (molecular weight (Mw) 80,000), Gelatin (Type A, 300 bloom from porcine skin), methacrylic anhydride (MA), 3-(Trimethoxysilyl)propyl methacrylate, Lipase from *Candida rugosa* (Type VII, ≥700 unit/mg solid), collagenase Type II from *Clostridium histolyticum* (0.5–5.0 FALGPA units/mg solid ≥125 CDU/mg solid), a glutaraldehyde solution (50 wt. in H$_2$O) and the photo-initiator 2-hydroxy-1-[4-(hydroxyethoxy) phenyl]−2-methyl-1-propanone (Irgacure 2959). Hexafluoroisopropan-2-ol (HFIP) was purchased from Oakwood Chemical. Dulbecco's phosphate buffered saline (DPBS), trypsin-EDTA (ethylenediaminetetraacetic acid), and penicillin–streptomycin were purchased from Gibco (MD, USA). Alpha-modified Eagle's medium (Alpha-MEM) was supplied by Invitrogen (Grand Island, NY, USA). HyClone characterized fetal bovine serum (FBS) and pre-cleaned microscope slides were obtained from Fisher Scientific (Waltham, MA, USA). 3-(4,5-dimethylthiazol-2-yl)−5-(3-carboxymethoxyphenyl)−2-(4-sulfophenyl)−2H-tetrazoli solution (MTS) was provided from Promega (USA).

**Procedure for the preparation of electrospun fibers**. The fibers were prepared by electrospinning a solution of respective polymer (10 wt%) or polymer mixture (Fig. 3a) in vials (10 mL) containing HFIP using 17 kV (Nanospinner Machine, Inovenso) as the positive voltage. Details are described in Supplementary Notes 1.

**Procedure for the preparation of PCLMA**. The electrospun PCL fibers were treated with plasma at 100 W for 5 min using the Anatech SP-100 Plasma System to graft highly reactive hydroxyl groups onto its surface (Fig. 2a). Subsequently, the hydroxylated PCL (PCL-OH) surface was immediately immersed in 3-(Trimethoxysilyl)propyl methacrylate in a glass container for 24 h at room temperature. Details are described in Supplementary Notes 2.

**General procedure for the preparation of GelMA**. GelMA was prepared by dissolving 10 g of gelatin type A from porcine skin in PBS (100 mL) at 50 °C[56]. Followed by the addition of 3 mL of methacrylate anhydride to the reaction vessel, the resulting mixture was kept stirring for 3 h at 50 °C. The material was dialyzed and lyophilized providing GelMA as a white solid. Details are described in Supplementary Notes 3.

**General procedure for the preparation of the PCLMA:GelMA blend**. Firstly, 1 g of PCLMA was dissolved in 10 mL of dichloromethane at 50 °C. Afterwards, the PCLMA solution was mixed using a magnetic stirrer for 24 h. Separately, GelMA (10 wt%) was dissolved in HFIP for 24 h. Then, the desired ratio of PCLMA and GelMA were combined and the solutions mixed under magnetic stirring for 30 min. Finally, the PCLMA:GelMA solution was electrospun as described in Supplementary Notes 1.

**Procedure for the first crosslinking—glutaraldehyde crosslinking**[57]. The fibers (10 mm length × 10 mm width × 1 mm depth) were crosslinked in a glutaraldehyde solution (10/500 mL ethanol) overnight. Subsequently, the reaction was quenched by the addition of a glycine solution (7.5 g/500 mL DI water). The fibers were rinsed several times with water and dried under a vacuum.

**Procedure for the second crosslinking—UV crosslinking**. The fibers from the previous step were immersed in an ethanol solution containing the photo-initiator for 2 h. The solution was prepared by dissolving 1.0 g of the photoinitiator (Irgacure 2559) in 10 mL ethanol in the absence of light and stirred until completely dissolved). Then, the fibers were crosslinked under UV light (365 nm, 6.9 mW/cm$^2$, using a Varian Cary 100 Bio UV–Visible spectrophotometer) at a 10 cm working distance for 10 min. Afterwards the electrospun fibers were washed with water several times and dried under a vacuum.

**Characterization**. A scanning electronic microscope (SEM) (Hitachi S-4800, 3 kV) was used to analyze the scaffold morphology. A thin gold layer was evaporated onto all scaffold surfaces before analysis to improve image acquisition. Fourier transform infrared spectrometry (FTIR, Spotlight-400, Perkin Elmer FTIR Imaging System) was used to analyze the chemical groups in/on the biomaterial before and after photocrosslinking under a controlled atmosphere and transmittance mode. Wettability analysis was performed by the contact angle (CA) measurements between the surface of the scaffold and a DI water drop monitored in dynamic mode, using a contact angle device (Krüss, Model DSA 100). Mechanical tests were performed before and after photo-crosslinking. Herein, an Instron Series 5944 testing system) was used in tension mode to perform mechanical tests on the scaffolds according to ASTM D 882-12 with some modification. $^1$H NMR analysis was performed by dissolving the PCL, PCLMA and 3-(Trimethoxysilyl)propyl methacrylate in deuterated chloroform (CDCl$_3$) (Sigma-Aldrich, St. Louis, MO) at a concentration of 10 mg/mL. The NMR experiment was performed at 25 °C on a Bruker Avance (400 MHz) spectrometer (Bruker, Harvard, MA). The transparency measurements were performed on a UV–Vis portable spectrometer UV 400 Ocean Optics, and sun light (400−1000 nm) was employed as a source of irradiation as depicted in Fig. 4d, ii−iv. Neutral density filters (5 and 60% of absorbance) were employed as references. The thermal behavior of the samples was analyzed using differential scanning calorimetry (DSC, Q2000, TA Instruments). Contact angle and surface energy measurements were performed on a goniometer (Krüss, Model DSA 100) operating under dynamic mode using water and diiodomethane. Details are described in Supplementary Notes 4−9.

**Cell biocompatibility in vitro**. To study the toxicity of the PCLMA:GelMA fibers, NIH/3T3 fibroblasts (American Type Culture Collection, VA, USA) were seeded in

the fibers and cultured in Dulbecco's Modified Eagle's Medium (DMEM) supplemented with 10% (v/v) fetal bovine serum (FBS) and 1% (v/v) penicillin-streptomycin at 37 °C with 5% $CO_2$. After 7 days of incubation, the viability of NIH/3T3 fibroblast cells in the fibers was analyzed using a LIVE/DEAD Viability/Cytotoxicity Kit (Thermo Fisher Scientific) following manufacturer's instructions. The number of live and dead cells on the samples (three samples per group) were counted using ImageJ software and the percentage of viable cells was calculated by dividing the number of live cells by the total cell number (Fig. 5b).

**Cell proliferation and spreading**. Spreading and proliferation of NIH/3T3 cells on the PCLMA:GelMA fibers were studied by immunocytochemical staining of F-actin using Alexa Fluor® 488 phalloidin (Molecular probes). Briefly, the cells were grown in the fibers under standard cell culture conditions as described above. At day 7, cells were fixed with 4% (v/v) paraformaldehyde for 20 min, permeabilized with 0.1% (w/v) Triton X-100 solution for 20 min and blocked with 1% (w/v) bovine serum albumin (BSA) for 1 h at room temperature. Then, Alexa Fluor® 488 phalloidin (1:100 dilution in 0.1% (w/v) BSA) was added to each sample and incubated for 1 h at room temperature to stain the filamentous actin (F-actin) cytoskeleton. After washing with 1× PBS for three times, the nucleus was stained with 4′,6-diamidino-2-phenylindole (DAPI) for 5 min at room temperature. Samples were imaged using a Zeiss AxioObserver inverted fluorescence microscope (Fig. 5c).

**In vitro degradation study**. The degradation test was performed by placing the fiber (1.0 × 1.0 cm) in 1.0 mL of PBS containing collagenase Type II (1 µg/mL) or 1.0 mL PBS containing lipase (1 µg/mL). Next, the samples were incubated at 37 °C and with 5% $CO_2$ in a humidified environment. Subsequently, after each selected time point, the fiber samples were removed from the exposure mixture and washed 1× carefully with DI water. Afterwards, the fibers were dried under vacuum at room temperature for 48 h and the dry weight recorded. Subsequently, the test continued with the same sample using a fresh solution. Testing for each fiber type was performed in triplicate. The degradation was calculated by following the equation: $(W_t - W_0)/W_t * 100\%$, where $W_t$ is the dry weight of the fiber at a specific time and $W_0$ is the initial dry weight of the fiber.

**In vivo biocompatibility study**. The implants were subcutaneously inserted on the dorsal region of the rats. In each animal, four incisions of approximately 8 mm were made along the back, two being on the left and two on the right side, using sterile fields, followed by divulsion with straight surgical scissors. The implants were randomly placed and the skin was sutured with a 4–0 nylon monofilament suture (ShalonVR). Animals were euthanized after 5 days with a lethal dose of anesthetic (Ketamine/Xylazine), and the biomaterials were harvested with surrounding tissue for histopathological analysis. Details are described in Supplementary Notes 10.

**Histopathological analysis**. The specimens were fixed in 10% buffered formalin (Merck, Darmstadt, Germany) for 24 h, followed by dehydration in a graded series of ethanol and embedding in paraffin. In the transverse axis to the implant, thin sections (5 µm) were prepared using a microtome (Leica Microsystems SP 1600, Nussloch, Germany). The specimens were stained with hematoxylin and eosin (H&E. stain, Merck) and examined using optical microscopy (Olympus Optical Co., Tokyo, Japan). The tissue response to subcutaneous implants was analyzed semi-quantitatively and a histological grading scale was used to evaluate the results (Table S2)[58-60]. Details are described in Supplementary Notes 11.

**In vivo hernia application**. Three specific pathogens free (SPF) B6/CBA F1 mice strain, 8 weeks old from the Multidisciplinary Center for Biological Research, Campinas, SP, Brazil, were employed for each group for the in vivo analysis of the various nanofibers. The animals were anesthetized with 100 mg/Kg ketamine and 10 mg/Kg xylazine IP before being submitted to surgery for the acquisition of a ventral abdominal hernia model[48], and repaired with the various PCL-based nanofibers. The procedure was adapted from the ventral abdominal hernia model by Suckow et al.[48]. A mechanical incision on the peritoneum was performed, causing a small hole of 0.5 cm of diameter, which was totally covered with the meshes produced and fixed with stitches using catgut sutures. Subsequently, the ventral abdominal wall was stitched using nylon. The animals were housed separately with water and food ad libitum in SPF condition. After 29 days, the animals were euthanized by overdosing with isoflurane, and the implanted material, as well as the surrounding tissue, were harvested. Hematoxylin and Eosin (H&E) were employed for H&E staining, and solutions of Hematoxylin, Acid Fuchsin and Methyl Blue were prepared for the Masson's Trichrome staining to distinguish the collagen fibers. For obtaining stretch fibers images, following the H&E staining samples were further studied under microscopy with a fluorescence filter (590-630 nm), and elastic fibers were detected by eosin fluorescence as suggested by Heo et al.[61]. For quantitative data analysis, a double-blind analysis was performed, i.e., the analyzer did not have data for features of the different tested nanofibers for biological analysis. A scoring table with subjective data was derived to appraise biological parameters observed using histological analysis. Specifically, immunological cells types, amounts and in vivo distributions; tissue recover and appearance

in the regions surrounding the implants; and blood vessels formation were investigated. Details are described in Supplementary Notes 12-14.

**Antibacterial study-colony count assays**. To investigate the antibacterial activity of the electrospun fibers, *Staphylococcus aureus* (*S. aureus*; ATCC 25923), *Methicillin-resistant S. aureus* (MRSA) (ATCC 43300), and *Pseudomonas aeruginosa* (*P. aeruginosa*; ATCC 25668) were used for the tests. The electrospun substrates (10 mm × 10 mm × 0.5 mm), which included PCL, PCLMA, GelMA-UV, PCLMA:GelMA (70:30) and PCLMA:GelMA (70:30)-UV, were inoculated each with 1 mL of the $10^4$ CFU/mL bacterial suspensions. Details are described in Supplementary Notes 15.

**Detection of ROS production**. The tests were performed using an intracellular fluorescence-based approach, reliant on the use of 2′,7′-dichlorodihydrofluorescein diacetate ($H_2$DCFDA), for the generalized detection of ROS. Details are described in Supplementary Notes 16.

**Scanning electron microscopy (SEM) analysis for bacterial adhesion**. The samples were exposed to the bacteria, and then the adherent cells were cross-linked using 2.5% of glutaraldehyde dispersed in 0.1 M cacodylate buffer as the primary fixative. Afterwards, the samples were fixed to mounts and loaded into the SEM. Details are described in Supplementary Notes 17.

**Statistics and reproducibility analysis**. All experiments were conducted in triplicate and were repeated at least three different times with analysis of variance (ANOVA) followed by a Tukey's multiple comparison test to determine statistical differences between mean values or by using ANOVA by the Kruskal–Wallis test with multiple comparisons. The populations from the analyzed scaffolds were obtained with normal distribution and independent to each experiment. The graphs were plotted by using GraphPad Prism 6 software. The in vivo hernia repair was performed with samples from each animal ($N = 3$ for each group) on 8-weeks-old B6/CBA F1 mice approved by the Ethical Committee for Laboratory Research Use of University of Campinas in SP- Brazil. Surgical procedures of the rats were conducted according to the Guiding Principles for the Use of Laboratory Animals. This study was approved by the Animal Care Committee guidelines of the São Carlos Federal University (protocol 8577280716). Ten male Wistar rats weighing 210–260 g and aged 8 weeks were used.

**Reporting summary**. Further information on research design is available in the Nature Research Reporting Summary linked to this article.

## Data availability
Data availability for the supporting data for the present study are available within this article or in the Supplementary Information file. Source data for the figures and bar graphs are provided in Supplementary Data 1-6. Any other data information is available from the authors on their reasonable request.

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

## Acknowledgements

This work was supported by the National Council for Scientific and Technological Development (CNPq, #310883/2020-2, #303752/2017-3 and #404683/2018-5 to AOL and #311531/2020-2, #304133/2017-5 and #424163/2016-0 to FRM) and Coordination for the Improvement of Higher Education Personnel (CAPES, #88881.120138/2016-01 to AOL and #88881.120221/2016-01 to FRM). Dr. Afewerki gratefully acknowledges financial support from the Sweden-America Foundation (The Family Mix Entrepreneur Foundation) and the Olle Engkvist Byggmästare Foundation. Dr. Harb has received financial support from Fundação de Amparo à Pesquisa do Estado de São Paulo (FAPESP, #2017/02899-1). The co-contributors from Northeastern University would like to thank Ebrahim Mostafavi of Northeastern University for providing advanced electrospinning training and Northeastern University for funding.
    Dedicated to the memory of Henok Afewerki (2017-11-16).

## Author contributions

S. A., G.U.R.-E., and A.O.L. designed the project and experiments and performed some of them. N.B. and S.V.H. performed the main in vitro experiments. D.W. and X.W. helped in the preparation of the fibers and antibacterial tests. M.M.M.de P. and M.A.F.C. performed the in vivo hernia experiments. S.M. performed the cell viability experiments. T.J.W. and F.R.M. designed the bactericidal experiments. C.R.T. performed the histological analysis. B.C.V. performed the transparency experiments. All the authors have contributed to the preparation of the final version of the paper and giving approval to the final paper.

## Competing interests

The inventors S.A., G.U.R.-E., and A.deO.L. have filed a patent application for the development of the disclosed technology through Brigham & Women´s Hospital, thus have commercial interest. All authors declare no competing interests.
