## [Peer Review File · Communications Biology]

Reviewers' comments:

Reviewer #1 (Remarks to the Author):

The authors have presented a very interested study and well-written manuscript.

The PT is been used in combination with electrospun fibers for over 10 years; examples:

1. Active screen plasma surface modification of polycaprolactone to improve cell attachment (2012): <https://doi.org/10.1002/jbm.b.31916>
2. Plasma Treatment of Electrospun PCL Random Nanofiber Meshes (NFMs) for Biological Property Improvement (2013): <https://doi.org/10.1002/jbm.a.34398>
3. Nanostructured surface of electrospun PCL/dECM fibres treated with oxygen plasma for tissue engineering (2016): <https://doi.org/10.1039/C6RA03840A>
4. Fabrication and Plasma Modification of Nanofibrous Tissue Engineering Scaffolds (2020): <https://doi.org/10.3390/nano10010119>
5. Effects of pre- and post-electrospinning plasma treatments on electrospun PCL nanofibers to improve cell interactions (2017): <https://doi.org/10.1088/1742-6596/841/1/012018>
6. Cold atmospheric plasma as a promising approach for gelatin immobilization on poly(ϵ -caprolactone) electrospun scaffolds (2019): <https://doi.org/10.1007/s40204-019-0111-z>

Moreover, PCL electrospun fibers with collagen or elastin for hernia applications with better orientation of fibers and with simpler method than the one reported this manuscript have previously reported, examples:

1. Nylon-6/chitosan core/shell antimicrobial nanofibers for the prevention of mesh-associated surgical site infection: <https://doi.org/10.1186/s12951-020-00602-9>
2. New surgical meshes with patterned nanofiber mats: <https://doi.org/10.1039/C9RA01917K>
3. Biocompatibility study of poly(vinyl alcohol)-based electrospun scaffold for hernia repair: <https://doi.org/10.3144/expresspolymlett.2018.58>
4. Fabrication and characterisation of drug-loaded electrospun polymeric nanofibers for controlled release in hernia repair: <http://www.sciencedirect.com/science/article/pii/S0378517316311644>

The authors need to use previous studies in order to explain the reason that their fabrication method is better as is not simpler.

Fabrication method

1. The fabrication method suggested in this manuscript and illustrated in Fig2, is complex for a mass production. Therefore, how this method can be scale-up and successfully be use in the future? This needs to be highlighted in the discussion.
2. The authors have not mentioned any concerns and suggestions for sterilisation. How these will be sterilise after manufacturing?

An expert opinion section will need to be included, covering issues and future directions, including FDA considerations.

Reviewer #2 (Remarks to the Author):

This study presents a novel biomaterial that has been developed to serve as a hernia repair device. The biomaterial combines several materials and fabrication techniques to leverage the strengths (and eliminate weaknesses) of each individual component. Hernias represent a common clinical problem and the prospect of repair by surgeons can be challenging given the wide range of biomaterials

available for use. An optimized biomaterial mesh could have high impact, thus this study is well motivated. The multidisciplinary research team accomplished a significant amount of experimental work on the new materials. Overall, the study was well done but a number of clarifications are needed and several suggestions are provided to improve the manuscript.

- Line 118: "Linea" should not be capitalized
- Lines 118-121: There appear to be a few mix-ups of parenthesis and brackets, which makes this hard to follow.
- Lines 118-121: A range of values have been reported for the mechanical properties of these abdominal tissues (see Deeken and Lake JMBBM 2017), which might be better to report than values from a single study. Either way, the values currently presented for the linea alba are labeled as "forces" but have units of Pascals which are stresses.
- Figure 1: some aspects of this figure are a bit sloppy and could be improved. E.g., the integrated strategy schematic is a bit silly and the text on each cartoon figure is difficult to read, the cloud shapes in the middle are a bit odd, and it's not clear how the PCLMA and PCLMA:GelMA relate to one another.
- Figure 2: very complex and lengthy fabrication process for the overall material. Would this be feasible for scale-up manufacturing for surgical use?
- Figure 3: in panel d, values were statistically compared to PCL100 and PCLMA100, yet there appear to be only one set of stars for each bar in the plots. So, not clear which statistical results are being shown.
- Lines 278-284: the true value of biomaterial transparency is unclear and perhaps a bit exaggerated by the authors; in addition, the measured transparency of the PCLMA fibers was outside the reference values used for scale calibration.
- Figure 4: in panel f, the authors report that data were only compared to PCL100 and PCLMA100; however, other bars appear to show comparisons between several other groups. What are these additional comparisons? And were these additional multiple comparisons accounted for in the statistical analysis?
- Figure 6 was described before Figure 5; order should be swapped.
- Figure 5: the four plots in panel d are not explained well; not clear what the scale of these values are and what they represent.
- Figures in general: a variety of details could be cleaned up to improve presentation. For example, sometimes the groups switch order of presentation or appear to be represented by different colors. Also, sometimes only a subset of the different biomaterials is evaluated by a specific assay but other times all groups are evaluated. It would be helpful to justify/explain why some groups were not considered for certain assays.
- Lines 292-294 and Figure 4: suddenly the authors start showing results for groups named "XXX-UV" (e.g., GalMA-UV), which were not shown previously. Are these new groups that were crosslinked with UV? But I thought previous groups (e.g., Figure 3) were also crosslinked via UV? This is a bit confusing.
- Figure 6: what is "tension fiber amount"? How are these fibers determined and what is their meaning? What information do these relay about the in vivo response to the various biomaterial meshes? The methods section seems to indicate that these are elastic fibers, however such fibers are not usually visible via eosin staining. Further verification would be needed to determine if these are actually elastic fibers. Either way, better interpretation of these fibers and their relevance is needed.
- Figure 6: the values shown in panel c are not clear; is a low or high value for each parameter "good" or "bad" in the context of biomaterial-host integration? Some of this information is reported in Table S2, but it would be clearer to include some of this information in Figure 6.
- General comment: throughout the paper, the authors frequently compare results to the PCL scaffold (e.g., response to bacterial infection). However, it appears that in many respects the PCL scaffolds are not a good option for hernia repair as the values/parameters/response are generally the poorest of all

the evaluated biomaterials. Thus, it would be better to focus group-group comparisons between the biomaterials that might be most likely to be successful in future work (e.g., PCLMA:GelMA 70:30 and other related groups). In other words, comparing a specific mesh to PCL typically isn't very helpful because PCL properties are often quite unfavorable.

Reviewer #3 (Remarks to the Author):

Thank you for a very detailed and interesting article. I have read it with huge interest and would like to wish you the best luck with succeeding. It would make lots of peoples lives much better.

I have few comments and have marked them by the page and line in the text, so you can find easier what am I referring to.

few general comments first. It would help if you have decided which hernias you are talking about, there are many types and many different repairs, each needing something slightly different. However the common principle is the same - to promote a production of more mechanically beneficial scar tissue. Unfortunately fibroblasts and collagen production is not the only thing we are aiming for. Most implants cause too much of it and it leads to a scar too stiff, foreign body sensation etc... Perhaps you could think how do you solve this problem with your implant.

Also it is not clear in what position of the abdominal wall you want to use your mesh. Please look at some pictures and make that decision, meshes for intraperitoneal placement are different.

few more comments here.

3/75 How do you mean high permeability?

5/116 Which type of hernia are you referring to? Incisional hernia definitely doesn't have recurrence rate 1,5% even with mesh.

5/125 You are talking only about midline ventral hernias then? Correct? Probably incisional, as primary ventral are in most cases quite small,

5/127 Also porcine and bovine products, synthetic many more materials too.

Fig. 1 - In which position do you aim to use your mesh? IPOM? or preperitoneal?

14/332 How would you explain that no cells attached to PCL when it did in other studies?

17/373 Abdominal muscle or fascia?

You may think I'm being difficult, but it is a great piece of engineering you have done and you need to fully understand the clinical importance of it, otherwise you are lowering your own chance of bringing a successful product to the market.

Referee expertise:

Referee #1: nanofibers in hernia repair

Referee #2: mechanical properties for biomaterials used for hernia repair

Referee #3: nanofibers hernia repair, in vivo study

Reviewers' comments:

Reviewer #1 (Remarks to the Author):

The authors have presented a very interested study and well-written manuscript.

The PT is been used in combination with electrospun fibers for over 10 years; examples:

1. Active screen plasma surface modification of polycaprolactone to improve cell attachment (2012): <https://doi.org/10.1002/jbm.b.31916>
2. Plasma Treatment of Electrospun PCL Random Nanofiber Meshes (NFMs) for Biological Property Improvement (2013): <https://doi.org/10.1002/jbm.a.34398>
3. Nanostructured surface of electrospun PCL/dECM fibres treated with oxygen plasma for tissue engineering (2016): <https://doi.org/10.1039/C6RA03840A>
4. Fabrication and Plasma Modification of Nanofibrous Tissue Engineering Scaffolds (2020): <https://doi.org/10.3390/nano10010119>
5. Effects of pre- and post-electrospinning plasma treatments on electrospun PCL nanofibers to improve cell interactions (2017): <https://doi.org/10.1088/1742-6596/841/1/012018>
6. Cold atmospheric plasma as a promising approach for gelatin immobilization on poly(ϵ -caprolactone) electrospun scaffolds (2019): <https://doi.org/10.1007/s40204-019-0111-z>

Moreover, PCL electrospun fibers with collagen or elastin for hernia applications with better orientation of fibers and with simpler method than the one reported this manuscript have previously reported, examples:

1. Nylon-6/chitosan core/shell antimicrobial nanofibers for the prevention of mesh-associated surgical site infection: <https://doi.org/10.1186/s12951-020-00602-9>
2. New surgical meshes with patterned nanofiber mats: <https://doi.org/10.1039/C9RA01917K>
3. Biocompatibility study of poly(vinyl alcohol)-based electrospun scaffold for hernia repair: <https://doi.org/10.3144/expresspolymlett.2018.58>
4. Fabrication and characterisation of drug-loaded electrospun polymeric nanofibers for controlled release in hernia repair: <http://www.sciencedirect.com/science/article/pii/S0378517316311644>

The authors need to use previous studies in order to explain the reason that their fabrication method is better as is not simpler.

Reply: We thank the reviewer for the valuable comments and suggestions. All the recommended references above have been added to the revised version of the manuscript and discussed highlighting the advantages of our integrated technology. The advantages with our disclosed fabrication approach are that it allows for the functionalization of various materials and as a proof of concept silylation was demonstrated however as we have shown in our previous work various functionalities and properties can be introduced by simple altering the functional groups added to the modification. Following sentences were added to the revised manuscript:

Nevertheless, the devised integrated strategy present in this work represents a broader and versatile concept than the presented previous reports and would allow the possibilities to functionalize various surfaces with a wide range of molecules, polymers and functional groups adaptable to extensive variety of properties and applications.²² However, to overcome some of the limitations with PT mentioned above and provide a solid fabrication process, PT can be merged with direct surface modification (DSM) leading to a more permanent modification.²²

Following sentences were also added in the hernia section:

*Various PCL-based electrospun fibers for hernia applications have previously been presented^{49,50} for instance, drug-loaded nanofibers with the antibiotic levofloxacin and antibacterial agent irligan for the prevention of potential bacterial infections,³⁸ nevertheless, this approach might promote the prevalence of multi-antibiotic resistant organisms.⁵¹ Moreover, PCL and polypropylene were combined for the generation of nanofibers with various orientations and patterns that demonstrated good mechanical properties and tunable cell morphology based on the patterning.⁵² However, in the presented technology the PCL is combined with GelMA which is known to promote various cellular and biological activities and tissue repair.²⁷ As *vide supra* highlighted the presented technology represent a wider and more versatile method compared to the previous studies, where it could allow facile tailoring of the engineered materials for instance the effortlessly addition of active ingredients or desired properties such as fluorophores etc. due to the important functional groups presented during the fabrication.²² References: 22. Afewerki, S. et al. Sustainable Design for the Direct Fabrication and Highly Versatile Functionalization of Nanocelluloses. *Global Challenges* **1700045**, 1700045-1700045, doi:10.1002/gch2.201700045 (2017). 27. Afewerki, S., Sheikhi, A., Kannan, S., Ahadian, S. & Khademhosseini, A. Gelatin-polysaccharide composite scaffolds for 3D cell culture and tissue engineering: Towards natural therapeutics. *Bioengineering & translational medicine* **4**, 96-115 (2019). 38. Hall Barrientos, I. J. et al. Fabrication and characterisation of drug-loaded electrospun polymeric nanofibers for controlled release in hernia repair. *International Journal of Pharmaceutics* **517**, 329-337, doi:<https://doi.org/10.1016/j.ijpharm.2016.12.022> (2017). 49. Molnar, K. et al. Biocompatibility study of poly(vinyl alcohol)-based electrospun scaffold for hernia repair. *Express Polym. Lett.* **12**, 676-687, doi:10.3144/expresspolymlett.2018.58 (2018). 50. Keirouz, A. et al. Nylon-6/chitosan core/shell antimicrobial nanofibers for the prevention of mesh-associated surgical site infection. *Journal of nanobiotechnology* **18**,*

1-17 (2020). 51. Afewerki, S. et al. *Advances in Dual Functional Antimicrobial and Osteoinductive Biomaterials for Orthopedic Applications*. *Nanomedicine: Nanotechnology, Biology and Medicine*, 102143, doi:<https://doi.org/10.1016/j.nano.2019.102143> (2019). 52. Liu, P., Chen, N., Jiang, J. & Wen, X. *New surgical meshes with patterned nanofiber mats*. *RSC advances* **9**, 17679-17690 (2019).

Fabrication method

1. The fabrication method suggested in this manuscript and illustrated in Fig2, is complex for a mass production. Therefore, how this method can be scale-up and successfully be use in the future? This needs to be highlighted in the discussion.

Reply: We thank the reviewer for the valuable comments. While the reviewer is right that the suggested fabrication looks lengthy when demonstrated in the figure, nevertheless the authors try to be as pedagogic as possible thus highlighting every step-in detail and its fundamental understanding of the chemistry in each fabrication process is important to address.

*In fact the functionalization/silylation process can be performed in shorter period of time for instance, as we have previously disclosed by the use of a catalyst (Ref. 17. Afewerki, S. et al. Sustainable Design for the Direct Fabrication and Highly Versatile Functionalization of Nanocelluloses. *Global Challenges* **1700045**, 1700045-1700045, doi:10.1002/gch2.201700045 (2017)). Moreover, the author is very experienced in converting lab technologies into industrial scale-up process and have made several of these translations, therefore, for a scaleup procedure the process will be made in one-pot or sequential manner, where all the steps would be integrated into one reaction batch*

or several and each process are performed consecutively. The only limiting step the author suspect in the disclosed technology for scale-up manufacturing is the plasma treatment, however, there are several industrial scale up process for plasma treatment. Moreover, there are several other alternative for introducing highly reactive oxygen speciece into the fibers that could be further investigated, for instance through treatment with hydrogen peroxide and sulphuric acid as presented by Cha et al. Ref. 16.

*Chaenyung, C. et al. Tailoring Hydrogel Adhesion to Polydimethylsiloxane Substrates Using Polysaccharide Glue. *Angewandte Chemie International Edition* **52**, 6949-6952, doi:doi:10.1002/anie.201302925 (2013).*

Following discussion have been added to the manuscript:

For a scale-up process of the disclosed strategy, the fabrication methodology would most

likely be employed in one-pot or sequential manner, where each steps of the fabrication process would be integrated into one reaction batch or several and each process are performed consecutively. Moroeover, the silylation steps would most likely be performed in the presence of a catalyst shortening the time significantly.

2. The authors have not mentioned any concerns and suggestions for sterilisation. How these will be sterilise after manufacturing?

We thank the reviewer for the important comment. There are several established procedures for the sterilization of PCL-based fibers that can be employed for the engineered fibers in the presented work such as β -irradiation, UV-irradiation or by the use of ethylene oxide. One strategy could be to try to link the sterilization procedure with the UV-crosslinking of the fibers (two birds with one stone). Following sentence with the respective references have been added to the revised version of the manuscript:

*Considering translational applications, the fibers could be sterilized by various established and known procedures such as β -irradiation, UV-irradiation or by the use of ethylene oxide. Ref. 41. Horakova, J. et al. Impact of Various Sterilization and Disinfection Techniques on Electrospun Poly- ϵ -caprolactone. ACS omega **5**, 8885-8892 (2020) and 42. de Cassan, D., Hoheisel, A. L., Glasmacher, B. & Menzel, H. Impact of sterilization by electron beam, gamma radiation and X-rays on electrospun poly-(ϵ -caprolactone) fiber mats. Journal of Materials Science: Materials in Medicine **30**, 42 (2019).*

An expert opinion section will need to be included, covering issues and future directions, including FDA considerations.

We thank the reviewer for the valuable comments and suggestions. This have been covered in the revised version of the manuscript. Following sentences have been added:

In the hernia section

*Considering regulatory aspects, since hernia support scaffolds are generally considered as medical devices and not as drugs, this will have considerable milder regulatory process.⁴⁹ Furthermore, the engineered nanofibers in this work comprise of PCL, which is an already United States Food and Drug Administration (FDA) approved material,³⁹ and according to FDA if the new device is equivalent to previous device it will be safe and effect as the counterpart, nevertheless supporting safety data may be demanded.⁵⁰ Generally bringing a medical devise to the market can take about 3–7 years, whilst about 12 years for drugs.⁵¹ References: 39. East, B. et al. A polypropylene mesh modified with poly- ϵ -caprolactone nanofibers in hernia repair: large animal experiment. International journal of nanomedicine **13**, 3129-3143, doi:10.2147/IJN.S159480 (2018). 49. Resnic, F. S. & Matheny, M. E. Medical Devices in the Real World. The New England journal of medicine **378**, 595 (2018). 50. Challoner, D. R. & Vodra, W. W. Medical devices and health—creating a new regulatory framework for moderate-risk devices. The New England journal of medicine **365**, 977-979 (2011). 51. Van Norman, G. A. Drugs, devices, and the FDA: part 2: an overview of approval processes: FDA approval of medical devices. JACC: Basic to Translational Science **1**, 277-287 (2016).*

In conclusion:

We envisage the engineered nanofibers will find its way to the clinics and solve challenging biomedical problems beside in hernia applications, moreover, we hope the integrated strategy will pave the way for the engineering of other tailor-made multifunctional biomaterials with the potential candidates in various applications.

Reviewer #2 (Remarks to the Author):

This study presents a novel biomaterial that has been developed to serve as a hernia repair device. The biomaterial combines several materials and fabrication techniques to leverage the strengths (and eliminate weaknesses) of each individual component. Hernias represent a common clinical problem and the prospect of repair by surgeons can be challenging given the wide range of biomaterials available for use. An optimized biomaterial mesh could have high impact, thus this study is well motivated. The multidisciplinary research team accomplished a significant amount of experimental work on the new materials. Overall, the study was well done but a number of clarifications are needed and several suggestions are provided to improve the manuscript.

- Line 118: “Linea” should not be capitalized
- Lines 118-121: There appear to a few mix-ups of parenthesis and brackets, which makes this hard to follow.
- Lines 118-121: A range of values have been reported for the mechanical properties of these abdominal tissues (see Deeken and Lake JMBBM 2017), which might be better to report than values from a single study. Either way, the values currently presented for the linea alba are labeled as “forces” but have units of Pascals which are stresses.

*Reply: We thank the reviewer for the valuable comments and suggestions. This have been corrected accordingly in the revised version of the manuscript. Moreover, the mechanical properties of the abdominal tissues we have presented in the manuscript are actually from the same report as suggested by the reviewer. Ref. 26: Deeken, C. R.; Lake, S. P. Mechanical properties of the abdominal wall and biomaterials utilized for hernia repair. *J. Mech. Behav. Biomed. Mater.* **2017**, *74*, 411–427.*

- Figure 1: some aspects of this figure are a bit sloppy and could be improved. E.g., the integrated strategy schematic is a bit silly and the text on each cartoon figure is difficult to read, the cloud shapes in the middle are a bit odd, and it’s not clear how the PCLMA and PCLMA:GelMA relate to one another.

Reply: We thank the reviewer for the valuable comments and suggestions. The figure 1 have been clarified and simplified in the revised version of the manuscript.

- Figure 2: very complex and lengthy fabrication process for the overall material. Would this be feasible for scale-up manufacturing for surgical use?

Reply: We thank the reviewer for the valuable comments. While the reviewer is right that it looks lengthy when demonstrated in the figure, nevertheless the authors try to be as pedagogic as possible thus highlighting every step-in detail and its fundamental understanding of the chemistry in each fabrication process is important to address.

In fact the functionalization/silylation process can be performed in shorter period of time for instance, as we have previously disclosed by the use of a catalyst (Ref. 22. Afewerki, S. et al. Sustainable Design for the Direct Fabrication and Highly Versatile

Functionalization of Nanocelluloses. Global Challenges **1700045**, 1700045-1700045, doi:10.1002/gch2.201700045 (2017)). The author is very experienced in converting lab technologies into industrial scale-up process and have made several of these translations, therefore, for a scaleup procedure the process will be made in one-pot or sequential manner, where all the steps would be integrated into a reaction batch or several and each process are performed consecutively. The only limiting step the author suspect in the disclosed technology for scale-up manufacturing is the plasma treatment, however, there are several industrial scale up process for plasma treatment. Moreover, there are several other alternative for introducing highly reactive oxygen speciee into the fibers that could be further investigated, for instance through treatment with hydrogen peroxide and sulphuric acid as presented by Cha et al. Ref. 16. Chaenyung, C. et al. Tailoring Hydrogel Adhesion to Polydimethylsiloxane Substrates Using Polysaccharide Glue. *Angewandte Chemie International Edition* **52**, 6949-6952, doi:doi:10.1002/anie.201302925 (2013).

Following discussion have been added to the manuscript:

For a scale-up process of the disclosed strategy, the fabrication methodology would most

likely be employed in one-pot or sequential manner, where each steps of the fabrication process would be integrated into one reaction batch or several and each process are performed consecutively. Moreover, the silylation steps would most likely be performed in the presence of a catalyst shortening the time significantly.

- Figure 3: in panel d, values were statistically compared to PCL100 and PCLMA100, yet there appear to be only one set of stars for each bar in the plots. So, not clear which statistical results are being shown.

Reply: We thank the reviewer for the valuable notice. It is a typos and it should be stated only PCL, since the various groups were compared to solely PCL group. This have been corrected in the revised version of the manuscript.

- Lines 278-284: the true value of biomaterial transparency is unclear and perhaps a bit exaggerated by the authors; in addition, the measured transparency of the PCLMA fibers was outside the reference values used for scale calibration.

Reply: We thank the reviewer for the valuable comments and notice. In fact, the transparency value of PCLMA fiber is a bit high. However, we used sun light as the light source and neutral density filters from THORLABS company (95 and 40% of transparency of visible light) as the reference for the calculations used to infer about the fiber's transparency. Besides, using the reviewer's advice/suggestion about the values, we reviewed the calculations (removing the baseline plateau). The values were recalculated for PCLMA giving up to 63% transparency, PCLMA:GelMA (70:30) and PCL = 19% and GelMA = 14% of transparency using the references normalized values of direct incident light without filter (100%) and the filters. We have remade the graphic of transmittance or transparency and inserted it in figure 4.

- Figure 4: in panel f, the authors report that data were only compared to PCL100 and PCLMA100; however, other bars appear to show comparisons between several other groups. What are these additional comparisons? And were these additional multiple comparisons accounted for in the statistical analysis?

Reply: We thank the reviewer for the valuable comments and observation. We apologies for this, it was a typo and should not be there. We have compared between all the groups and multiple comparison was accounted in the statistical analysis as stated in the manuscript.

- Figure 6 was described before Figure 5; order should be swapped.

Reply: We thank the reviewer for the valuable comment and observation. This have been corrected in the revised version of the manuscript.

- Figure 5: the four plots in panel d are not explained well; not clear what the scale of these values are and what they represent.

Reply: We thank the reviewer for the valuable comments. This have been corrected in the revised version of the manuscript. The scales are based on the histological grade scoring explained in Table S2, this have been highlighted in the legend text of the figure and the y-axis have been highlighted with this, moreover figure titles have been added to each plot explaining what they represent.

- Figures in general: a variety of details could be cleaned up to improve presentation. For example, sometimes the groups switch order of presentation or appear to be represented by different colors. Also, sometimes only a subset of the different biomaterials is evaluated by a specific assay but other times all groups are evaluated. It would be helpful to justify/explain why some groups were not considered for certain assays.

Reply: We thank the reviewer for the valuable comments and suggestions. These have been addressed in the revised version of the manuscript and the authors have tried to improve the presentations.

- Lines 292-294 and Figure 4: suddenly the authors start showing results for groups named “XXX-UV” (e.g., GalMA-UV), which were not shown previously. Are these new groups that were crosslinked with UV? But I thought previous groups (e.g., Figure 3) were also crosslinked via UV? This is a bit confusing.

Reply: We thank the reviewer for the valuable comments and observations. This have been corrected in the revised version of the manuscript. Figure 3 have been assigned with XXX-UV, since the crosslinked group are crosslinked with UV. However, in the revised version of the manuscript this confusing has been addressed.

- Figure 6: what is “tension fiber amount”? How are these fibers determined and what is

their meaning? What information do these relay about the in vivo response to the various biomaterial meshes? The methods section seems to indicate that these are elastic fibers, however such fibers are not usually visible via eosin staining. Further verification would be needed to determine if these are actually elastic fibers. Either way, better interpretation of these fibers and their relevance is needed.

*Reply: We thank the reviewer for the valuable comments and observations. We have added a description of what we mean with tension fibers for the first time this expression has appeared in the revised version of the manuscript. The meaning for tension fibers is a combined fiber on the tissue under stretching strengths, as stress fibers (composed of actin and non-muscle myosin II) as well as elastic fibers, which are valuable for the new tissue recovery. Actin is widely known to have an affinity to eosin, and when it is under stress, it can show thick fibers. Besides, the elastic fibers, which are also fibers involved in stretching stress, also has strong, but no-specific, the affinity of eosin as supported by previous literature (Heo, Y. S. & Song, H. J. Characterizing cutaneous elastic fibers by eosin fluorescence detected by fluorescence microscopy. *Annals of dermatology* **23**, 44-52 (2011); and Vilamaior, P.S.L., Suzigan, S., Carvalho, H.F., Taboga, S.R. Structural characterization and distribution of elastic system fibers in the human prostate and some prostatic lesions. *Braz. J. Morphol. Sci.*, 20, 101—107 (2003)). This was valuable because we could quantify intensity by eosin fluorescence and make structural analysis of these fibers (but not specify) at the recovering tissue. Since this tissue is a continued tissue under stretching action, the meshes that support these tension fibers formations had a better rating.*

- Figure 6: the values shown in panel c are not clear; is a low or high value for each parameter “good” or “bad” in the context of biomaterial-host integration? Some of this information is reported in Table S2, but it would be clearer to include some of this information in Figure 6.

Reply: We thank the reviewer for the valuable comments and observations. This has been clarified at the figure in the revised version of the manuscript, where we have indicated that higher number represents improved rating.

- General comment: throughout the paper, the authors frequently compare results to the PCL scaffold (e.g., response to bacterial infection). However, it appears that in many respects the PCL scaffolds are not a good option for hernia repair as the values/parameters/response are generally the poorest of all the evaluated biomaterials. Thus, it would be better to focus group-group comparisons between the biomaterials that might be most likely to be successful in future work (e.g., PCLMA:GelMA 70:30 and other related groups). In other words, comparing a specific mesh to PCL typically isn't very helpful because PCL properties are often quite unfavorable.

Reply: We thank the reviewer for the valuable comments and suggestions. The idea with the paper was a proof of concept presentation on the concept with the integrated strategy comprising the combination of electrospinning, plasma treatment and subsequent direct surface modification for the engineering the multifunctional nanofibers. This strategy also allow for tailoring of various biomaterials, as a proof of concept applications PCL was selected due to its limitations mentioned by the reviewer

and by the authors throughout the manuscript. Therefore, it was important for us to highlight the superior performance of the engineered PCL-based nanofibers compared to solely PCL (the initial control material). Furthermore, throughout the manuscript we also tried to compare and highlight the differences between PCL:GelMA (70:30) and PCL:GelMA (70:30)-UV to depict the impact from crosslinking on the nanofibers. However, despite this was the focus throughout the manuscript several groups were compared and discussed between each other.

Reviewer #3 (Remarks to the Author):

Thank you for a very detailed and interesting article. I have read it with huge interest and would like to wish you the best luck with succeeding. It would make lots of peoples lifes much better.

I have few comments and have marked them by the page and line in the text, so you can find easier what am I refering to.

few general comments first. It would help if you have decided which hernias you are talking about, there are many types and many different repairs, each needing something slightly different. However the common principle is the same - to promote a production of more mechanically beneficial scar tissue. Unfortunately fibroblasts and collagen production is not the only thing we are aiming for. Most implants cause too much of it and it leads to a scar too stiff, foreign body sensation etc... Perhaps you could think how do you solve this problem with your implant.

Reply: We thank the reviewer for the valuable comments and suggestions. The authors agree with the reviewer; more studies must be performed to better understand in detail the mesh's fitness and how it will be at the end; better scar tissue, best welfare for the patient during and after recovery, and without a doubt, it is an issue we have to address in the future. We are at the beginning. Here we have presented the material's development with different compositions to follow each mesh's physical, chemical, and biological properties. Regarding biological characteristics, we addressed cytotoxicity, biocompatibility, the first contact mesh-to-tissue, and tissue recovering into the live organisms. That work was the first screening that indeed shows us improvements and disadvantages of each material that can be used for a more profound practical approach focused in the pre-clinical assays.

Also it is not clear in what position of the abdominal wall you want to use your mesh. Please look at some pictures and make that decision, meshes for intraperieotneal placememtn are different.

Reply: We thank the reviewer for the valuable comments and observations. At the present work, we used meshes stitched over the peritoneal wall to cover a hernia caused by mechanical incision. It was to test the material properties and their behavior in contact with alive organisms; under different mechanical and biological conditions. The chosen material for pre-clinical approaches will have a better scope, and specific surgery skills to get results closer to clinical reality.

few more comments here.

3/75 How do you mean high permeability?

Reply: We thank the reviewer for the valuable comment. When we talk about permeability, we indicate the three-dimensional porous and microstructure of the scaffold and this will highly impact the mass transport and oxygen permeability. Several reports have disclosed the importance of high permeability PCL scaffold for various tissue engineering and biomedical applications. A high permeability of a PCL scaffold for instance the value $4.1 \times 10^{-7} \text{ m}^4/\text{N}\cdot\text{s}$ is considered a high permeability as reported by Hollister et al. (Tissue Engineering: Part A, 2011, 17, 1831.). The group demonstrated that higher permeability PCL scaffold promotes higher amounts of bone ingrowth, bone penetration, with increased blood vessel infiltration in implanted on a mouse model. Overall, the study demonstrated that higher permeable PCL scaffold would be preferable for in vivo bone regeneration. This reference have been added to the revised version of the manuscript providing the reader further insight into the importance of permeability of PCL scaffolds.

5/116 Which tyoe of hernia are you referereng to? Incisional hernia deffinitely doesnt have recurrence rate 1,5% even with mesh.

*Reply: We thank the reviewer for the valuable comment. According to the previous work they talk about the recurrence rate of 1.5% in incisional hernia comparison to repair without implant. see references: 25. Poussier, M. et al. A review of available prosthetic material for abdominal wall repair. Journal of visceral surgery **150**, 52-59, doi:10.1016/j.jviscsurg.2012.10.002 (2013) and Évaluation des implants de réfection de paroi, de suspension et d'enveloppement en chirurgie digestive et dans les indications spécifiques à la chirurgie pédiatrique. HAS; 2008. p. 1—14. 2.*

5/125 You are talking only about midline ventra hernias then? Correct? Probably incisional, as primary ventral are in most cases quite small,

Reply: We thank the reviewer for the valuable comment. Yes, the previous report talks about incisional hernia see references: 1. Song, C., Alijani, A., Frank, T., Hanna, G., Cuschieri, A., 2006. Elasticity of the living abdominal wall in laparoscopic surgery. J. Biomech. 39 (3), 587–591 and 2. Song, C., Alijani, A., Frank, T., Hanna, G.B., Cuschieri, A., 2006. Mechanical properties of the human abdominal wall measured in vivo during insufflation for laparoscopic surgery. Surg. Endosc. 20 (6), 987–990.

5/127 Also porcine and bovine prodcuts, synthetic many more materials too.

Reply: We thank the reviewer for the valuable comment. We have added these examples and many more synthetic materials in the revised version of the manuscript.

Fig. 1 - In which position do you aim to use your mesh? IPOM? or preperitoneal?

Reply: We thank the reviewer for the valuable comment. At the present work, we used meshes stitched over the peritoneal wall to cover a hernia caused by mechanical incision. It was to test the material properties and their behavior in contact with alive organisms; under different mechanical and biological conditions.

14/332 How would you explain that no cells attached to PCL when it did in other studies?

Reply: We thank the reviewer for the valuable comment. Due to the high hydrophobic nature of the PCL fibers the cells were not able to attached to the scaffold, similar observation has been observed by several reports were very low attachment have been detected. Following sentence was added to the revised manuscript:

*No cells attachment was observed to the PCL nanofibers due to its highly hydrophobic nature.¹² (12. De Paula, M. M. M. et al. Understanding the impact of crosslinked PCL/PEG/GelMA electrospun nanofibers on bactericidal activity. PLOS ONE **13**, e0209386, doi:10.1371/journal.pone.0209386 (2018)).*

17/373 Abdominal muscle or fascia?

Reply: We thank the reviewer for the valuable comment, it should be abdominal muscle.

You may think Im being difficult, but it is a great piece if engineering you have done and you need to fully understand the clinical importance of it, otherwise you are lowering your own chance of bringing a successfull product to the market.

REVIEWERS' COMMENTS:

Reviewer #1 (Remarks to the Author):

The authors have addressed all the comments of this reviewer and revised the manuscript.

Reviewer #2 (Remarks to the Author):

The authors have responded to most of my comments and the manuscript is improved. However, I am still very confused about the "tension fiber amount" parameter and the authors' response only confused me more. They mention "stretching strengths", which makes no sense, and a better rating from these "tension fibers", which I don't understand. Please explain clearly what you are visualizing and measuring with this parameter and how it relates to the mechanics of the tissue. Also, can you distinguish between actin/myosin stress fibers and elastic fibers or are these all lumped together? Wouldn't this matter?

Reviewer #3 (Remarks to the Author):

Thank you for explanation and changes made.
I have few more, sorry for that.

Wording in the abstract is little bit unusual. "the abnormal exodus of tissue and/or organs termed a hernia is a great
33 clinical challenge that currently needs surgery for recovery"

5/120 A hernia never heals without surgical repair, and in the worst case can lead to infection, sepsis and possible death of some parts of the gastrointestinal tract (from a ventral abdominal hernia).

Hernia doesn't lead to this, its incarcerated hernia.. when not operated in time, and not only ventral but any.

122 These kinds of complications are normally repaired employing an implant that reinforces the abdominal wall, promotes the repair and decreases recurrence rate (<1.5%).

Which complications? 1,5% is a recurrence rate of a very good hernia centre for a groin hernia. Definitely not incisional or primary ventral. Please decide which hernias are you referring to and then correct the data.

143 not prompt any inflammatory reaction

Inflammatory reaction is part of healing.. Did you mean chronic?

430 Are you referring to the k510 rule? It refers to a condition you already have a PCL mesh on the market.. There is PCL used and approved, but no nanofibrous PCL mesh.

657 and repaired with the various PCL-based nanofibers

It would be clearer if you said how you did it, in an inlay position, defect wasn't closed, peritoneum stayed untouched? suture material?